# Early human impact on lake cyanobacteria revealed by a Holocene record of sedimentary ancient DNA

Ebuka Canisius Nwosu [1✉], Achim Brauer[2,3], Marie-Eve Monchamp[4,5], Sylvia Pinkerneil[2], Alexander Bartholomäus[1], Martin Theuerkauf[6], Jens-Peter Schmidt[7], Kathleen R. Stoof-Leichsenring [8], Theresa Wietelmann[8], Jerome Kaiser [9], Dirk Wagner[1,3] & Susanne Liebner [1,10]

Sedimentary DNA-based studies revealed the effects of human activity on lake cyanobacteria communities over the last centuries, yet we continue to lack information over longer time-scales. Here, we apply high-resolution molecular analyses on sedimentary ancient DNA to reconstruct the history of cyanobacteria throughout the Holocene in a lake in north-eastern Germany. We find a substantial increase in cyanobacteria abundance coinciding with deforestation during the early Bronze Age around 4000 years ago, suggesting increased nutrient supply to the lake by local communities settling on the lakeshore. The next substantial human-driven increase in cyanobacteria abundance occurred only about a century ago due to intensified agricultural fertilisation which caused the dominance of potentially toxic taxa (e.g., *Aphanizomenon*). Our study provides evidence that humans began to locally impact lake ecology much earlier than previously assumed. Consequently, managing aquatic systems today requires awareness of the legacy of human influence dating back potentially several millennia.

[1] GFZ German Research Centre for Geosciences, Section Geomicrobiology, 14473 Potsdam, Germany. [2] GFZ German Research Centre for Geosciences, Section Climate Dynamics and Landscape Evolution, 14473 Potsdam, Germany. [3] Institute of Geosciences, University of Potsdam, 14476 Potsdam, Germany. [4] Department of Biology, McGill University, Montreal, QC H3A 1B1, Canada. [5] Groupe de Recherche Interuniversitaire en Limnologie (GRIL), Montreal, QC H3A 1B1, Canada. [6] Institute of Botany and Landscape Ecology, University of Greifswald, 17489 Greifswald, Germany. [7] State Authority for Culture and Preservation of Monuments of Mecklenburg and Western Pomerania, State Archaeology, 19055 Schwerin, Germany. [8] Polar Terrestrial Environmental System, Alfred Wegener Institute Helmholtz Centre for Polar and Marine Research, 14473 Potsdam, Germany. [9] Leibniz Institute for Baltic Sea Research, 18119 Rostock-Warnemünde, Germany. [10] Institute of Biochemistry and Biology, University of Potsdam, 14476 Potsdam, Germany. ✉email: enwosu@gfz-potsdam.de

Climatic and anthropogenic activities are known to shape aquatic communities, but the role of climate change versus human pressure on lake biodiversity remains difficult to disentangle. Within the last century, anthropogenic drivers have strongly influenced the abundance and diversity of cyanobacteria[1–3], often leading to massive blooms that reduce water quality[4] and threaten various ecosystem functions[2,5]. In the worst cases, cyanobacteria blooms are dominated by taxa that produce toxins that are harmful to zooplankton, birds, and mammals, including humans[6,7]. However, due to the paucity of long-term data, the dynamics of cyanobacteria on millennial timescales remain poorly determined.

Lake sediments are a repository for organic and inorganic material that is produced within the lake or transported from its catchment and act as natural archives for regional climatic and environmental history[8]. Lake sedimentary archives thus provide evidence of natural and anthropogenic-driven ecosystem changes[9] recorded in various proxy data. Early cyanobacteria paleolimnological studies commonly used fossilised remains (akinetes, cysts)[10–13], pigments[14,15], and biomarkers[16,17] preserved in sediments as proxies for inferring past changes. However, not all cyanobacteria leave sufficient identifiable remains in sediments to allow for a high degree of taxonomic differentiation as distinctive species (e.g., picocyanobacteria[18]), and the presence of highly labile conjugated double bonds makes some of their biomarkers unstable (e.g., the carotenoids zeaxanthin and echinenone)[19]. Therefore, the potential to reconstruct past cyanobacteria variability based on classical fossil-based proxies is limited. Within the last decade, a variety of DNA-based paleolimnological methods have emerged as novel and complementary proxies for use in reconstructing past cyanobacteria variability at improved accuracy[9], including physiologically dormant, active, and inactive organisms buried in sediments[20]. Major advantages of DNA-based approaches include their high phylogenetic resolution and their reliable coverage of the entire cyanobacteria phylum preserved in sediment archives[18,21,22]. Recent sedaDNA studies have linked changes in freshwater cyanobacteria structure and diversity to variations in temperature, precipitation, and eutrophication as well as increased lake circulation and dust input[21–23]. However, most sedaDNA studies on cyanobacteria report data spanning only the last few centuries[21,22] up to a maximum of the last two millennia[23]. Consequently, our knowledge is restricted to a period that has been influenced by both climatic changes and intensive human impact, and cyanobacteria dynamics over large periods of the Holocene remain understudied. This knowledge deficit limits our understanding of whether the influence of earlier, prehistoric cultures on lakes led to changes in cyanobacteria dynamics compared with the undisturbed, natural conditions before the presence of the first human settlements in the region.

The present-day mesotrophic Lake Tiefer See located in north-eastern Germany is ideal for studying potential early human impacts on lake cyanobacteria. Lake Tiefer See was formed at the end of the last glaciation as part of the Klocksin lake chain and has a maximum depth of 62 m (Fig. 1a–c). The lake's sediments have been studied under various aspects[24], including climatic and environmental history based on sedimentological and geochemical proxies[25,26], as well as vegetation openness reconstructions[27]. The main characteristic of the sediment record is the alternation of varved (annually laminated) and non-varved intervals, which indicates that the lake is sensitive to climatic and environmental changes[25]. Lake Tiefer See is an integrated part of the TERENO long-term monitoring programme[28], and seasonal sedimentation processes as well as their drivers are well known[29].

The objective of the present study is to reconstruct the dynamics of cyanobacteria throughout the entire Holocene in the well-studied deep temperate Lake Tiefer See and to discuss how far back in time anthropogenic factors began to influence cyanobacteria community composition, abundance, and diversity. We use a variety of high-resolution molecular approaches including shotgun (metagenome) sequencing, amplicon sequence variants analysis (ASVs; a proxy for different species and variants within the same species), quantitative polymerase chain reaction (qPCR) assays, and cyanobacteria lipid biomarker (7-methyl-heptadecane) analyses, and we combine these approaches with the reconstruction of vegetation openness (land-cover changes inferred from pollen data[27]) in order to investigate changes in cyanobacteria over the past ~11,400 years. Lake Tiefer See was selected for our study because the lake's sedimentation has been monitored since 2012[29] and the processes of cyanobacteria DNA transport from water to the burial in sediments have been studied in detail[30], thereby providing a robust framework for our data interpretation. Settlements by sedentary farmers and ranchers within the Lake Tiefer See catchment and wider region appeared during the Bronze Age (ca. 3940–3100 cal. a BP)[31,32]. During the Iron Age the lake area was largely depopulated. This period was then followed by intermittent human occupation during ca. 2700–2200, 990–895 (CE 960–1055), and since 750 cal. a BP (CE 1200), with settlements preferentially built on the banks of water.

## Results

**Core sampling and chronology.** The two new overlapping core sequences obtained for this study extend an earlier composite record from Lake Tiefer See down to ca. 1180 cm. The onset of the Holocene is clearly reflected by a distinct shift towards organic lacustrine sedimentation and further confirmed by a change in pollen assemblage. The lowermost part of the record is largely homogeneous followed by a characteristic alternation of homogeneous and varved sediment intervals including different varve types as earlier described[25]. The period of recent varve formation commenced about a century ago favoured by anthropogenic eutrophication[33]. The sediments are mainly composed of three components, biochemically precipitated calcite, amorphous and particulate organic matter, as well as diatom frustules. Owing to the lack of a major inflow, detrital minerogenic components are rare and mostly consist of scattered silt-sized quartz and carbonate grains. The published age model for the last 6000 years is based on varve counting, radiocarbon dating and tephrochronology[25], whereas the older part is based on radiocarbon dating, tephrochronology and biostratigraphy (see Materials and Methods).

**Lake environmental variables.** The total organic carbon (TOC) content was the highest in sediment samples from the mid-Holocene (mean 17.5 wt.%; ca. 8470–4070 cal. a BP) and the lowest in the basal early-Holocene sediment sample (1.4 wt.%; ca. 11,340 cal. a BP), as well as in samples from the Little Ice Age (mean: 7.2 wt.%; 717–96 cal. a BP; ca. CE 1230–1850). A correspondence between TOC content and varve preservation was observed[25]. The pollen-based vegetation reconstruction shows repeated changes in the proportion of open vegetation such as grasslands or arable land and forests around Lake Tiefer See (Fig. 2). The lowest vegetation openness values (<20%)—which reflected a largely forested landscape—were observed from ca. 11,340–4070 cal. a BP and during short periods between ca. 3970 and 810 cal. a BP. The highest openness values (>35%) were reconstructed in the samples spanning the time periods of ca. 3960–3360 cal. a BP, ca. 2800–2100 cal. a BP, and since ca. 880 cal. a BP (ca. CE 1070), which reflect periods of enhanced human settlement[27].

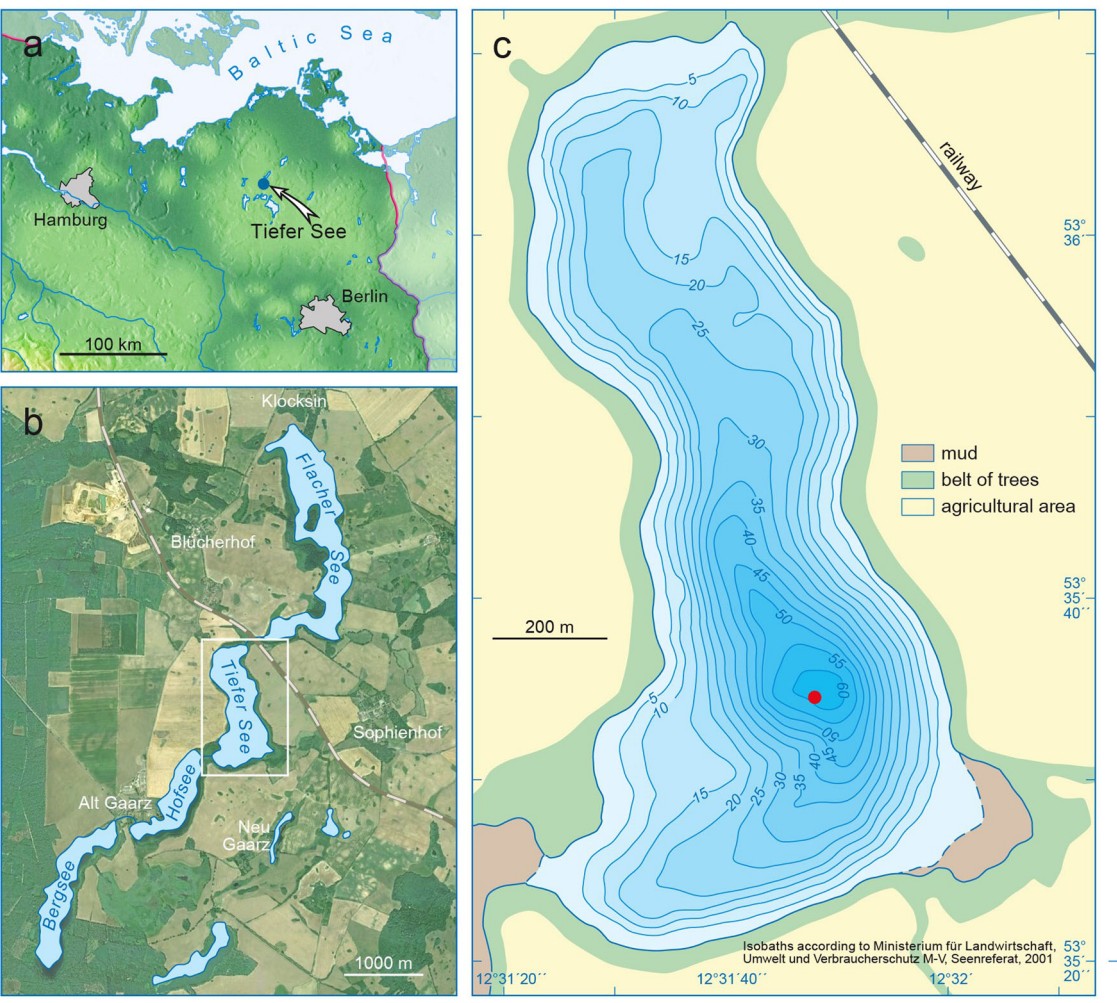

**Fig. 1 Study site. a** Location of Lake Tiefer See in the southern Baltic lowlands (blue dot), **b** Lake Tiefer See within the Klocksin lake chain, and **c** topographic map of Lake Tiefer See showing the coring site (red dot).

**SedaDNA concentration and cyanobacteria abundance.** Over-all, the sedaDNA concentration normalised over sediment wet weight and the total organic carbon showed comparable long-term trends (Fig. 2 and Supplementary Table 1). The main finding is a distinctly higher sedaDNA concentration since ca. 3960 cal. a BP ($n = 57$, mean: 1.4 µg g$^{-1}$ wet sediment and 0.14 µg TOC$^{-1}$ %) than between ca 11,340 and ca. 4070 cal. a BP ($n = 17$, mean: 0.2 µg %$^{-1}$ wet sediment and 0.02 µg TOC$^{-1}$%). Cyanobacteria gene copies were most abundant in the topmost sediment sample (CE 2000; $2.3 \times 10^5$ copies ng$^{-1}$ DNA). Other peaks in cyanobacteria gene copies were detected in samples dated to ca. 810 cal. a BP ($8.4 \times 10^4$ copies ng$^{-1}$ DNA) and ca. 1900 cal. a BP ($2.1 \times 10^5$ copies ng$^{-1}$ DNA) following a pro-gressive increase since ca. 3970 cal. a BP. The lowest number of cyanobacteria gene copies were detected from ca. 11,340 to ca. 4070 cal. a BP, where values ranged between $10^2$ and $10^3$ gene copies ng$^{-1}$ DNA.

The robustness and quality of cyanobacteria sedaDNA were validated via cyanobacteria lipid biomarker analysis (7-methyl-heptadecane, Supplementary Table 2), clone libraries created from selected qPCR amplification products for total cyanobac-teria (~350 bp), and long cyanobacteria amplicons (~1500 bp) using long-fragment primers (Methods, Supplementary Table 3). The results confirm that cyanobacteria DNA fragments are well preserved along the entire sediment core (11 m, Supplementary Note 2).

**Cyanobacteria ASV analysis.** The analysis of the 100 most abundant cyanobacteria ASVs revealed that *Cyanobium* was the most abundant (>50%) genus in the sediment samples (Fig. 3). ASVs assigned to potential bloom-forming and toxin-producing genera such as *Aphanizomenon, Microcystis, Dolichospermum,* and *Planktothrix* were recorded throughout the sediment core from ca. 10,820 cal. a BP. Beginning in the 1950s, *Aphanizome-non* became the most abundant (25–50%) potentially toxin-producing genus. In sediment samples spanning from ca. 2610–660 cal. a BP, *Planktothrix* was the most abundant (~25%) potentially toxin-producing genus. Specifically, *Planktothrix* abundance peaked in samples from ca. 1650–1580 cal. a BP. The peak in abundance was further confirmed by calculating ratios of the relative abundance of *Planktothrix* ASVs to the total cyano-bacteria abundance (qPCR). The proportion of *Planktothrix* in the cyanobacteria population confirmed the above-mentioned observations (Supplementary Fig. 3). Altogether, cyanobacteria ASV (taxonomic) richness was generally higher in more recent sediment layers (42–169; beginning ca. CE 1870; Fig. 2; Supple-mentary Table 2) compared with sediment layers from ca. 11,340–5070 cal. a BP (7–40). Between ca. 4070 and 100 cal. a BP, cyanobacteria ASV richness ranged from 14–90.

Non-metric multidimensional scaling (NMDS) revealed three significantly different temporal clusters of cyanobacteria com-munity composition at the ASV level. The youngest cluster covers only the last 150 years, whereas the two older clusters comprise

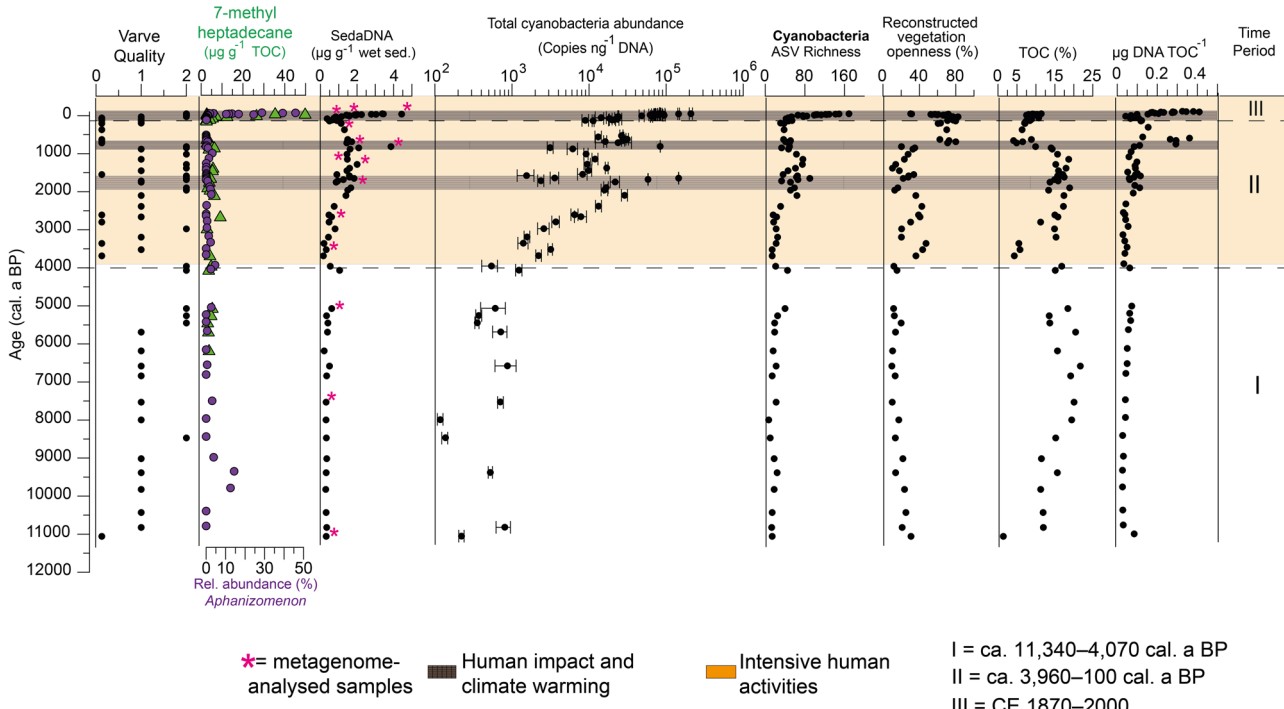

**Fig. 2 Sediment chronology with sedimentological parameters, sedaDNA content, information on cyanobacteria abundance and alpha diversity, and selected lake paleoenvironmental records.** Sedimentological analysis: Varve quality (where 0 = non-varved, 1 = poorly varved, and 2 = varved). Abundance of ASVs assigned to *Aphanizomenon* relative to total cyanobacterial ASVs (in %) based on amplicon sequencing shown for the entire Holocene (purple dots) compared to 7-methylheptadecane normalised to total organic carbon lipid biomarker shown up to ca. 6500 cal. a BP (green triangles; μg g$^{-1}$ TOC). Microbiological analysis: SedaDNA concentration was normalised to sediment weight (μg DNA g$^{-1}$ sed; magenta-coloured stars indicate metagenome-analysed samples) and to total organic carbon content (μg DNA TOC$^{-1}$); total cyanobacteria abundance was determined via quantitative PCR; error bars give the standard deviations for three independent amplifications; cyanobacteria taxonomic richness was retrieved from amplicon sequencing and ASV analysis. Geochemical analyses: TOC contents. Pollen-based reconstructed vegetation openness data[27]. Dashed black lines demarcate three significantly different temporal clusters identified by non-metric multidimensional scaling (I) ca. 11,340–4070 cal. a BP, (II) ca. 3960–100 cal. a BP, and (III) ca. CE 1870–2000 (see Fig. 4). Orange-coloured sections highlight a period with phases of intensive human activities around the study site, while dark-brown grids highlight peaks in cyanobacteria abundance coinciding with warming periods.

several millennia (Fig. 4). A conservative one-way permutational analysis of variance (PerMANOVA) and a subsequent pairwise test confirmed that the three different clusters are significantly different from one another (*p*-value = 0.0001; *F* = 7; Supplementary Table 5). A Spearman rank-based correlation further determined that cyanobacteria abundance (qPCR) positively correlated with vegetation openness (*n* = 74, *p*-value = 0.003, $R_S$ = 0.5; Supplementary Table 6), while the correlation between vegetation openness and cyanobacteria ASV richness was also positive but non-significant (*n* = 74, *p*-value = 0.1, $R_S$ = 0.26). Both sediment weight and total organic carbon-normalised DNA contents showed significant positive correlations to cyanobacteria ASV richness (*n* = 74, *p*-value = 0.003, $R_S$ > 0.8) and to cyanobacteria abundance (*n* = 74, *p*-value = 0.003, $R_S$ = 0.7). The positive correlation between species richness and VQ was significant (*n* = 74, *p*-value = 0.003, $R_S$ = 0.5) but the positive correlation was not significant between cyanobacteria abundance and varve quality (*n* = 74, *p*-value = 0.1, $R_S$ = 0.2).

**Metagenomic analysis.** A shotgun DNA sequence analysis was used to assess the presence of the cyanobacteria functional genes that were responsible for nitrogen fixation (*nif* gene family*)* and microcystin synthesis (*mcy* gene family) in a subset of 14 samples (Fig. 5a, b), as well as to verify the taxonomic assignment from amplicon sequencing (Supplementary Fig. 4). The abundance of *nif* genes distinctly increased for the first time in samples dated to between ca. 3520 and 530 cal. a BP, and ranged from 0.012–0.02%

of all cyanobacteria genes. The level of *nif* abundance was highest (0.05–0.20%) in samples from CE 1940, 1970, and 1990. In samples where *nif* abundance was recorded, a relatively high abundance of the diazotroph *Aphanizomenon* was also observed via amplicon sequencing (Fig. 3). The *mcy* genes were recorded for the first time in a sample dated to ca. 1900 cal. a BP. Similar to *nif*, *mcy* was most abundant in recent sediment strata (CE 1940 and 1990).

Taxonomic mapping based on shotgun sequencing of the 16S rRNA gene confirmed the observed changes based on ASV data, with minor differences in the relative abundance of cyanobacteria groups (Supplementary Fig. 4). For example, the family *Cyanobiaceae* (which contains the ubiquitous genus *Cyanobium*) was the most abundant in all the shotgun-analysed samples similar to the amplicon sequencing for the same samples, in which *Cyanobium* was also most abundant. Other abundant families based on shotgun data were *Phormidiaceae*, *Nostocaceae*, *Microcystaceae*, and *Synechococcaceae*. The low overall number of shotgun sequence reads for cyanobacteria in samples dated to ca. 5070, 7530, and 11,340 cal. a BP (<8,000,000 reads) compared with those in the other samples (>22,000,000) may explain the absence of *nif* and *mcy* functional genes in these samples. The number of reads (114,185) obtained for the basal sample dated to ca. 11,340 cal. a BP was too low for further downstream analyses. Analysis of cyanobacteria ancient DNA (aDNA) mean fragment length distribution using mapDamage2.0 show reduction in the number of reads and in DNA fragment size with increasing depth (Fig. 6).

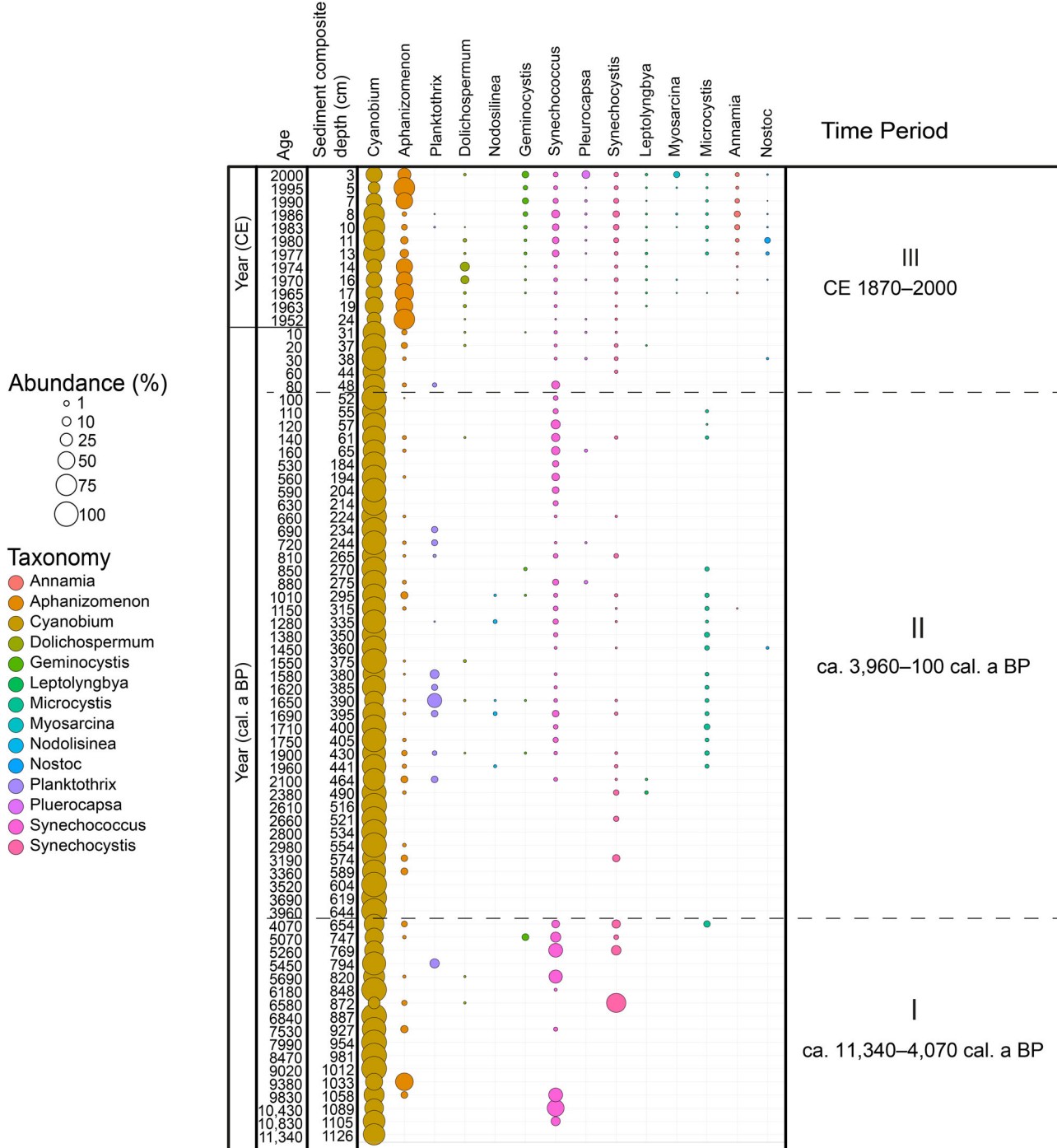

**Fig. 3 Cyanobacteria taxonomic distribution.** Bubble plot showing the variation of cyanobacteria community composition at the genus level (cutoff >0.1% relative abundance, the tiny dots represent ~0.1% abundance) spanning over 11,000 years. Dashed black lines demarcate the three significantly different temporal clusters identified by non-metric multidimensional scaling (see Fig. 4).

## Discussion

Our sedaDNA reconstruction spanning the last ca. 11,340 years provides the first record of cyanobacteria DNA for the entire Holocene from the freshwater Lake Tiefer See. The most abundant recovered taxon was *Cyanobium*, which together with *Synechococcus* (Picocyanobacteria) are generally more abundant (often by several orders of magnitude) than large bloom-forming taxa in lakes[34]. From the nearby Baltic Sea sediments, cyanobacteria glycolipids produced by heterocystous nitrogen-fixing cyanobacteria such as *Aphanizomenon* have been traced to the

sea's earlier freshwater phase at ca. 7200 cal. a BP[35]. We now provide the first evidence that cyanobacteria -including potentially toxic taxa like *Aphanizomenon*- have been present in Lake Tiefer See since the early Holocene (Fig. 3). This finding concurs with previous studies suggesting that cyanobacteria are pioneer organisms in aquatic environments[36], meaning that they are among the initial colonisers of newly formed ecosystems. Surprisingly, we detected the first substantial increase in cyanobacteria abundance as early as 3960 cal. a BP, which indicate a major increase in human-induced clearcutting and land openness

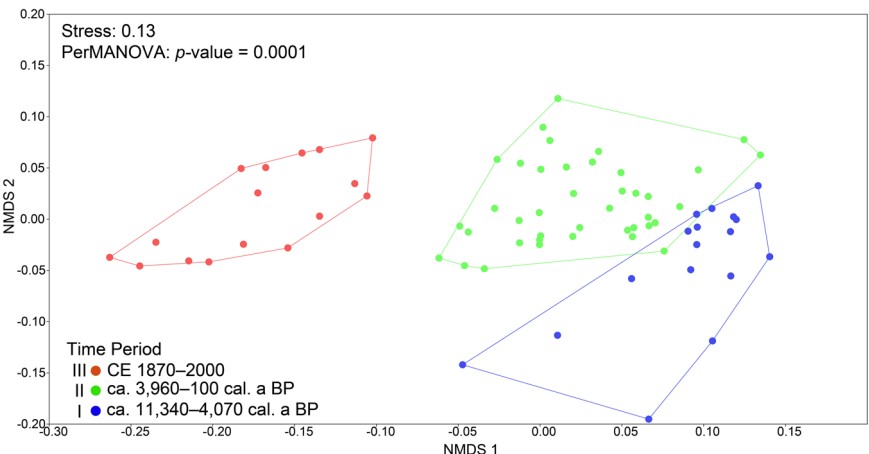

**Fig. 4 Cyanobacteria Holocene beta-diversity analysis based on amplicon data.** A non-metric multidimensional scaling (NMDS) visualisation of cyanobacteria community beta diversity based on amplicon sequence data using Bray–Curtis dissimilarity. The dots represent the sample depths analysed.

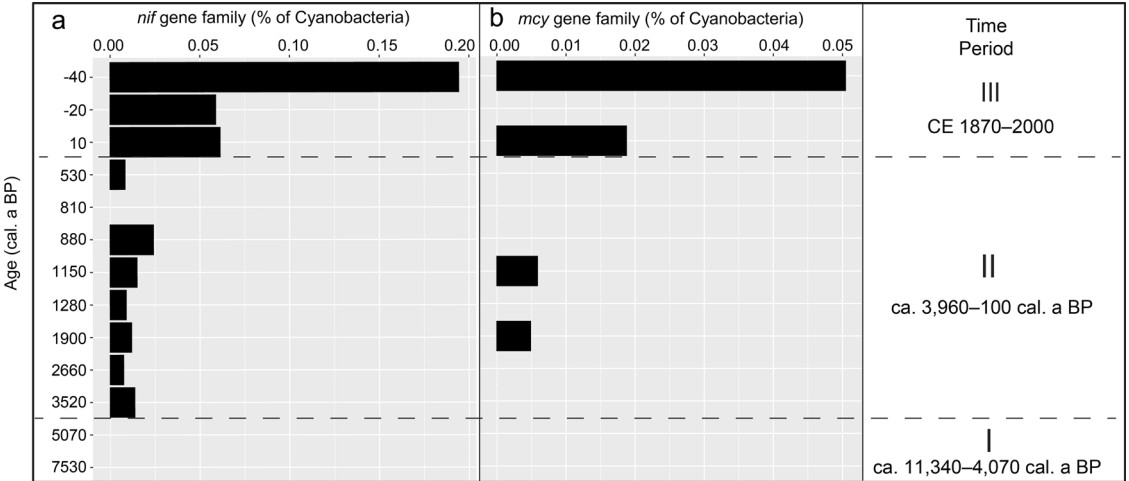

**Fig. 5 Cyanobacteria gene families based on shotgun data.** Bar plots of key cyanobacteria-affiliated gene families involved in **a** nitrogen fixation (*nif*) and **b** microcystin (*mcy*) biosynthesis. The normalised gene counts are expressed in percentage over all cyanobacteria counts for the selected sediment samples spanning over 11,000 years. In samples with no bars, no cyanobacteria-affiliated *nif* (or *mcy*) genes were detected. Dashed black lines demarcate the three significantly different temporal clusters identified by non-metric multidimensional scaling (see Fig. 4).

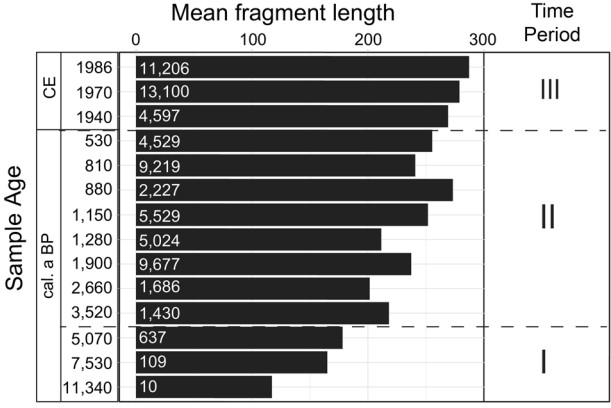

**Fig. 6 Cyanobacteria ancient DNA reads from shotgun data.** Bar plot showing cyanobacteria mean fragment length distribution from shotgun-analysed samples. Total cyanobacteria reads from each analysed depth is indicated in each bar. Dashed black lines demarcate the three significantly different temporal clusters identified by non-metric multidimensional scaling (see Fig. 4).

(from 12–36%) between ca. 3960 and 3690 cal. a BP. This is confirmed by pollen data (Fig. 2). These increases in cyanobacteria abundances and land openness coincided with a time for which archaeological evidence points towards Bronze Age settlements around Lake Tiefer See[31,37–39] (Supplementary Fig. 1). Archaeological findings—such as a burial mound, swords, a lanyard, socket axe, and a ring dated to the Bronze Age—have been recovered from Lake Tiefer See and neighbouring Flacher See (Fig. 1b)[31,37,38], suggesting intensive human activities in the area. It seems likely that these early settlers built walled burial houses and two-aisled buildings near the lake in which handicraft activities—such as pottery or bronze foundry—were practiced[32], thus, their activity and life-style likely had a direct impact on the lake ecosystems. These anthropogenic activities are consistent with archaeology reports on human settlements in the region during the Bronze Age[32,40]. The availability of water for trade and private use played an important role in the selection of the settlement site, especially as there was arable land in the vicinity of Lake Tiefer See on which barley and emmer were likely grown[41]. Therefore, we hypothesise that the substantial increase in cyanobacteria abundance despite no obvious change in sedaDNA

content was driven by intensified human activities such as deforestation and land use[27] for farming within the catchment.

Studies based on akinetes have been used to trace eutrophication during the Holocene in various lakes (e.g., lakes Gosciaz[11], Wielkopolska[42], Bruszczewo[43], Lanoto'o[13], and Nussbaumersee[12]) revealing varying effects of anthropogenic pressures on cyanobacteria. For example, in Lake Lanoto'o (Uplou, Samoa), no changes in the algae and cyanobacteria communities were observed with the arrival of humans in the lake region ca. 2800 years ago[13]. This is because Lake Lanoto'o is in a remote higher-elevated area of the island and thus insulated against anthropogenic-induced ecosystem change. In Lake Nussbaumersee (Canton of Thurgau, Switzerland), an increased abundance of akinetes produced by cyanobacteria taxa *Aphanizomenon*, *Anabaena*, and *Gloeotrichia* was coincident with human settlement periods in the region[12]. Additionally, it was argued that climatic changes and related changes in the levels of Lake Nussbaumersee (max depth 8.7 m), may have been drivers of the recorded cyanobacterial blooms[12]. Lake Nussbaumersee was cutoff from the other basins during low lake-level periods, thus the resulting eutrophication was a mixture of human impact and lower lake levels, which also enhanced human-impact effects[12]. Similar lake-level arguments have been used to explain past cyanobacteria blooms in Lake Moon (max. depth 6.5 m, China) reconstructed from sedaDNA[23]. It is thought that lower lake levels caused by arid climates between 1900 and 560 cal. a BP led to increased concentrations of sediment recycled nutrients in the water column in Lake Moon, which ultimately promoted cyanobacteria blooms. However, in contrast to lakes Nussbaumersee and Moon, higher lake levels due to land-cover changes (deforestation) occurring during settlement phases were recorded for Lake Tiefer See for the time periods ca. 4000–3000, ~2500, ~1800, ~1200, ~1000 and 600 cal. a BP[44]. This strongly suggests environmental forcing from human impact rather than internal nutrient cycles as the main driver of cyanobacteria abundance in the deep Lake Tiefer See during the Bronze Age (4000–3000 cal. a BP). Furthermore, it was concluded that the resilience of Lake Nussbaumersee was responsible for the decline in the akinete abundance of the cyanobacterial taxa *Aphanizomenon*, *Anabaena*, and *Gloeotrichia* following abandonment. In contrast, the observed shift to higher total cyanobacteria abundance since the Bronze Age (Fig. 2) did not reverse to pre-human settlement periods in Lake Tiefer See (ca. 11,400–4000 cal. a BP) following abandonment, even when clearcutting and land openness substantially decreased (e.g., down to 10% at ca. 1380 cal. a BP; Fig. 2). This finding suggests a long-lasting legacy effect of anthropogenic eutrophication on Lake Tiefer See cyanobacteria community dynamics during the Bronze Age. The sources of excessive nutrient addition into Lake Tiefer See were likely from animal and human excrement, land clearing itself, and/or use of fires for preparing land for agriculture (ash contains phosphorus). These nutrients were accumulated in the lake after inundation due to the lack of a major outflow[24] and likely stored in the sediments. It should, however, also be noted that our study quantified total cyanobacteria (qPCR) compared to the study in Lake Nussbaumersee which is based on the relative abundances of identifiable remains of three cyanobacteria taxa.

In addition to cyanobacteria abundance, cyanobacteria beta-diversity also changed substantially beginning with the Bronze Age as shown by NMDS (Fig. 4). Additionally, *nif* genes affiliated with cyanobacteria were detected for the first time at ca. 3520 cal. a BP (Fig. 5a). We consider the detectable increase in *nif* gene affiliated with cyanobacteria indicative of their increase in demand for nitrogen likely as a result of an increasing population. Either growth of cyanobacteria populations resulted in an increasing demand for nitrogen or the abundance of *nif* genes simply

increased along with the increase in population size. However, unlike abundance (Fig. 2), beta-diversity did not change as much in the Bronze Age. It was only from ca. 2610 cal. a BP that beta-diversity increased substantially (Fig. 3), perhaps due to increasing air temperature after the Homeric solar minimum[45,46], or another unknown factor. An effective break in cyanobacterial community composition that began during the Bronze Age in response to human activity is further supported by the substantial increase in the cyanobacteria lipid biomarker 7-methylheptadecane content at ca. 2660 cal. a BP (up to 6.8 µg g TOC$^{-1}$; Supplementary Table 2; Supplementary Fig. 6).

The next major and mainly anthropogenic-induced increase in cyanobacteria abundance occurred at the beginning of industrialisation at ca. CE 1870 (Figs. 2 and 3). Since this period, intensified human impact via mechanised farming, the use of artificial fertilisers[33], and deforestation[27] have again resulted in a substantial increase in cyanobacteria abundance that clearly differs from that of the Bronze Age. Simultaneously, the onset of industrialisation coincides with a marked shift in cyanobacteria community composition, which began to display a pronounced dominance of the eutrophication-associated and potentially toxic *Aphanizomenon* (Fig. 3). Based on this study, this change in cyanobacteria abundance and taxonomic composition is unprecedented in the previous eleven millennia. It is likely that climate warming during recent decades and anthropogenic eutrophication via agriculture (that peaked with pig farming in the 1970s[22,47]) with its attendant nutrient loading both contributed to these observed changes in cyanobacteria community dynamics (which has also been found in other deep temperate lakes[48,49]). It is, however, difficult to differentiate climatic from anthropogenic eutrophication effects. Similarly, we cannot unambiguously relate the peak in cyanobacteria abundance dated to ca. 1690 cal. a BP (which corresponds to the Roman Iron Age[39,50]) to human impact because warm climate at the time[45,46] (Supplementary Fig. 2) coincided with settlement structures in the immediate vicinity of the lake, as indicated by archaeological evidence (Supplementary Fig. 1).

Essentially, our results show that the two most substantial shifts in cyanobacteria community dynamics throughout the Holocene were primarily caused by human activities. In contrast, the early to mid-Holocene—which was characterised by warmer summers[51,52], during climate optima at both ca. 9500 and 7000 cal. a BP in the Northern Hemisphere[52]—showed no detectable changes in cyanobacteria community dynamics (Figs. 2 and 3). This finding strongly suggests that human impact has been the dominant driving factor behind cyanobacteria community dynamics in Lake Tiefer See during the last eleven millennia. With climate warming in recent decades (meteorological data from DWD Station Schwerin CE 1890–2016; 80km to the west of Lake Tiefer See), increasing water temperatures may have played an increasing role for cyanobacteria dynamics. Our findings are supported by reports that nutrients mainly control cyanobacteria abundance in oligotrophic lakes while temperature may be a more important driver in mesotrophic lakes[53]. Based on phosphorus concentrations, the present-day Lake Tiefer See is mesotrophic[24], however, we assume more oligotrophic conditions during the early to mid-Holocene, with human absence. This assumption could explain the relative stability in cyanobacteria community dynamics between ca. 11,400 and 5000 cal. a BP, and the observed substantial changes in cyanobacteria community dynamics with the intensification of human settlements and agricultural activities ca. 4000 years ago. Human impacts since then have likely been the trigger responsible for the even more recent peaks in cyanobacteria abundance due to legacy effects. Even if climate warming played an additional role during the Roman Iron Age and the Medieval period (as it most likely also

does in recent times), our data strongly suggest that the later peaks in cyanobacteria abundance were only possible because of the initial human impact during the Bronze Age. Our assumption is that the increase in cyanobacteria abundance dating back to the Bronze Age initiated the lasting build-up of a community in which changes in community composition and function became visible. This assumption is apparent with the first detection of the potentially toxic *microcystis* synthesis (*mcy*) gene at ca. 1900 cal. a BP (Fig. 5b). *Planktothrix* abundance based on amplicon data had also reached high values at ca. 1900 cal. a BP (Fig. 3 and Supplementary Fig. 3). *Planktothrix* is one of the freshwater cyanobacteria genera that produce microcystin[54], and even the main contemporary cyanotoxin producer in lakes of northern Germany[55]. The detection of *mcy* genes since ca. 1900 cal. a BP could indicate the onset of toxic cyanobacterial bloom events, and a substantial increase in toxigenic taxa since CE 1940 (e.g., *Microcystis*; Figs. 3 and 5b) although genetic information alone cannot inform the toxicity of cyanobacteria populations[56]. *Mcy* genes may not have been detected before 1900 cal. a BP either because the gene was not present at all, or only at too low concentrations to be detected using a shotgun sequencing approach.

Our multi-proxy study based on cyanobacteria DNA and biomarker analyses provides evidence of the long-term preservation of lake cyanobacteria DNA in lake sediments dating back to ca. 11,400 years. A foremost challenge to sedaDNA data interpretation is the current limited knowledge of taphonomic processes affecting DNA during the transport into sediments, and preservation of DNA in sediments under prevailing environmental conditions[20]. We have previously shown that the mainly anoxic cold water–sediment interface in Lake Tiefer See promotes DNA preservation after incorporation[22,30]. Here, we report a continuous increase in cyanobacteria abundance (from $5.4E + 02$ to $6.5E + 03$ copies ng$^{-1}$ DNA between ca. 3960 and 2660 cal. a BP) despite consistently low sedaDNA concentration ($<0.8\ \mu g\ g^{-1}$ wet sediment; Fig. 2; Supplementary Tables 1 and 2). This finding suggests a reliable signal because the copy numbers were normalised over the amount of DNA in each sample (Supplementary Fig. 4). The shotgun cyanobacteria data also confirms that the oldest samples have hardly any cyanobacteria reads and that a substantial pool of cyanobacteria has developed around ca. 1900 cal. a BP (Supplementary Fig. 4a), which is consistent with the first Holocene peak in cyanobacteria abundance around ca. 1900 cal. a BP (qPCR; $2.1 \times 10^5$ copies ng$^{-1}$ DNA).

Our data show that sediment characteristics such as varve quality had no impact on the reconstructed cyanobacteria abundance (qPCR; *p*-value = 0.1; Fig. 2; Supplementary Fig. 5; Supplementary Table 5). However, varve quality positively correlated with the number of reconstructed cyanobacteria ASV (richness), this means that varved sediments had a higher number of different cyanobacteria taxa, and suggest that varve preservation promotes DNA preservation[9]. Therefore, while varve quality can have an impact on the number of reconstructed cyanobacteria taxa (alpha diversity), their community clustering patterns (beta diversity) are not impacted by varve quality. Additionally, the trend observed in the cyanobacteria lipid biomarker 7-methylheptadecane reconstructed over the last 6000 years from Lake Tiefer See supports the DNA-based results from this study and suggests that there was no substantial degradation of ancient cyanobacteria DNA (Fig. 2). Even at the species level, the relative abundance of *Aphanizomenon* —which is also known to produce 7-methylheptadecane[17,57,58]— shows similar variability to 7-methylheptadecane contents over time in some samples, which is in line with previous evidence[22]. For example, the relative abundance of both *Aphanizomenon* and 7-methylheptadecane contents increased slightly at ca. 3960 cal. a BP (Supplementary Table 2). Other peaks in 7-methylheptadecane that do not correspond to *Aphanizomenon* may have

been produced by other cyanobacteria producers of 7-methylheptadecane[57,58].

Although contamination in sedaDNA studies cannot be completely ruled out[20], we took precautions to minimise the risks (see Methods). The DNA extraction and library preparation kits used were not optimised for aDNA, which might have led to the loss of some short and degraded DNA fragments, especially in the oldest sediment samples. That, and the use of relatively large targets (>350 bp) for amplicon and qPCR, atypical in aDNA studies[59], may have contributed to limiting the detection of the cyanobacterial signal in the early-Holocene sediments, which is a potential bias of our study. Nevertheless, the general decrease in cyanobacteria mean fragment length distribution with increasing depth in the shotgun data indicates that the cyanobacteria dynamics at the bottom of the core were indeed ancient (Fig. 6). The four mean fragment length outliers dated to 880-, 1150-, 1900-, and 3520 cal. a BP may have been due to better preservation of cyanobacteria aDNA, that is, better protection of aDNA since they originate from well-varved sediments[9], or different DNA fragmentation in library preparation protocol. We could not obtain any cyanobacteria aDNA misincorporation plots with the typical "smile" patterns using mapDamage[60] or pyDamage[61]. This may have been because the protocols of the commercial library preparation kits were modified and improved by our sequencing partner Eurofins (who cannot provide us with details due to property rights). We assume the library preparation step may have included Uracil Specific Excision Reagent (USER) treatment. The USER treatment inadvertently removes deamination signals by excising uracils expected on library hairpin adapters—uracils, which also happen to be present in ancient DNA. We would expect this to then bias a library away from ancient molecules, leaving no C-T or G-A misincorporations that can be observed downstream with bioinformatic damage pattern tools. Another reason could be low coverage, that is, the low number of cyanobacteria reads in the shotgun samples, which was between 10 in the deepest (ca. 11,400 cal. a BP) and 13,000 in the topmost (CE 1986) sediment (Fig. 6). This sequencing depth is too low to cover all available cyanobacteria genomes from NCBI (1,119,256,536 reads) or the *Cyanobium* genome (3,342,364 reads). To overcome this limitation, future shotgun-based studies on Holocene cyanobacteria aDNA could use sedaDNA hybridisation capture techniques like the biotinylated RNA baits[62,63] to enrich for cyanobacteria aDNA. It needs to be considered though that the biotinylated RNA baits were optimised and tested against environmental and marine eukaryotic sedaDNA, which contributes a much smaller proportion to total sedaDNA than prokaryotic sedaDNA. Therefore, we hypothesise that combining hybridisation technique, a modified sequencing library preparation without uracil-excision, and deeper shotgun sequencing could increase the recovery of ancient cyanobacteria diversity.

Despite the possible limitations discussed above the results from our different sedaDNA approaches (qPCR, amplicon, and shotgun sequencing) show similar temporal trends to the cyanobacteria biomarker analyses. Our key finding—sustained increase in cyanobacteria abundance at ca. 4000 cal. a BP—is clearly supported by pollen-based reconstructed vegetation openness and archaeological findings in and around Lake Tiefer See. Based on these data we exclude contamination by modern DNA.

Our study allowed us to compare cyanobacteria community dynamics over time periods that preceded recent human influence since the industrialization by almost four millennia. Using high-resolution molecular analysis on sedaDNA we generated a robust data set that enhanced our understanding of the dynamics of cyanobacteria community dynamics since the early Holocene

to the present. In addition to the known anthropogenic-induced increase in lake cyanobacteria in the last 150 years (which also clearly showed up in our data), we found striking evidence for a distinct increase in cyanobacteria abundance at around 4000 years ago, which is likely related to the rise of settlements around Lake Tiefer See and the associated agricultural activity during the early Bronze Age. This period marks an unprecedented shift in cyanobacteria abundance and has since not declined to pre-settlement levels, even in times when human impact in the lake catchment had decreased. The impact of human activity on lake cyanobacteria during the Bronze Age triggered later changes in the abundance and composition of cyanobacteria through the build-up of a much larger population than that observed during the preceding millennia without human activity. This means that human activity began to locally impact lake ecology much earlier than in the previously assumed last 150 years. Creating awareness of the long-lasting legacy impact of human influence on lake cyanobacteria populations is important when managing aquatic systems that face imminent dangers from expanding potentially toxic species.

## Methods

**Lake hydrology.** The source of water of Lake Tiefer See is mainly rainfall and to some extent subsurface flow and groundwater. Lake Tiefer See is part of a chain of lakes but the connection to the northern Lake Flacher See was piped during railway constructions from CE 1884–1886 and is in modern times mostly dry[33]. The present connection to Lake Hofsee in the South is through a wetland and is very shallow (Fig. 1b). According to the reconstructed water level changes throughout the Holocene, lake levels were always at least 1–2 m below the range of present-day fluctuations until ca. 2800 cal. a BP so that Lake Tiefer See could not have been connected to Lake Hofsee during the Bronze Age.

**Core sampling and chronology.** In May 2019, two new overlapping sequences of 2-m long sediment cores were obtained from borehole H (6 cores: TSK19_H1-H6) and borehole K (5 cores: TSK19_K1-K5) in the deepest part of Lake Tiefer See (62 m; latitude 53°35′36″N, longitude 12°31′46″E; Fig. 1a–c), using a 90-mm UWITEC piston corer. These boreholes are located nearby previous cores[25]. The cores were opened, documented and stored in cold room in the same way as previous cores[25] and could be precisely correlated to the published composite core by distinct marker layers. Furthermore, the new core sequences were complete so that two minor core gaps below 750 cm sediment depth in the previous composite profile[25] could be bridged facilitating a complete Holocene sediment record. The age model of the sediment composite profile back to 6000 years ago has been established by integrating varve counting, tephrochronology, and radiocarbon dating[25]. The older part of the age model between 6000 cal. a BP and the onset of the Holocene is based on 14 new and one published[25] radiocarbon dates on terrestrial macro-remains, one visible tephra layer (Saksunarvatn) and 3 cryptotephra horizons (Lairg-B, Askja-S, Hässeldalen (all published[25,64])) and precisely correlated from the previous cores to the new ones. The onset of the Holocene is further confirmed by biostratigraphy. Ages are given either as calibrated years before present 1950 (cal. a BP) and/or common era (CE). Evidence of human settlement in the vicinity of Lake Tiefer See during the Bronze Age, the Pre-Roman Iron Age and the Roman Iron Age are based on archaeological findings (Supplementary Fig. 1).

**Past vegetation openness reconstruction and total organic-carbon analyses.** The past vegetation openness was reconstructed from pollen percentage data using the REVEALS model[65]. The model was applied with the REVEALS function in R from the DISQOVER package with the Lagrangian stochastic model for pollen dispersal and the PPE.MV2015 data set[27]. Total organic carbon (TOC) content for the last 6000 years has been published previously[25]. For the present study, we additionally measured samples covering ca. 6000 and 11,340 cal. a BP using the same methods in the Climate Dynamics and Landscape Evolution isotope laboratory at the German Research Centre for Geosciences (GFZ). Briefly, 1-cm bulk samples were freeze-dried, ground, homogenized, and analysed using an EA Isolink elemental analyser coupled to a DELTA V Advantage Isotope Ratio Mass Spectrometer (ThermoFisher Scientific)[25]. TOC was examined by in-situ decalcification of ~1 mg of sediment wrapped in Ag capsules by treating it with 3% and 20% HCl for 3 h and drying it at 75 °C. The calibration was performed based on an elemental Urea and checked with an internal soil reference sample (Boden3, HEKATECH). Replicate analyses of the standards resulted in a reproducibility of 0.1 wt.% for TOC. The TOC content is expressed as percent of dry weight (wt.%).

**Sedimentary ancient DNA analysis.** A total of seventy-four sediment layers were sampled from the middle of the cores. To avoid contamination, the top layers were

scraped off after core opening with a sterile razor and the uncontaminated anoxic part below was put into sterile 15-mL falcon tubes using sterile spatulas and stored at −80 °C until analysis. Cross-contamination of the sediments was avoided by processing recent and older samples at different times. The weight of the sediment used for each extraction was ~250 mg. Four DNA extractions were performed per sample and the DNA extracts were pooled in a 2 mL Eppendorf Tube after extraction. These combined DNA extracts were used for the qPCR assay and amplicon sequencing. The total DNA extraction was done using the PowerSoil DNA Isolation Kit (Qiagen) at the German Research Centre for Geosciences (GFZ) in Potsdam, as previously described[22]. DNA extractions were completed in batches of seven samples with the addition of one negative control (extraction blanks without sediment) in each batch and following strict protocols in order to reduce contamination from foreign DNA[21]. The genomic DNA extracts were checked with agarose gel electrophoresis and Tape Station (older samples ca. 3190 cal. a BP; Agilent 2200 D5000 ScreenTape) to verify that the fragment lengths were sufficient for downstream analyses. The sedaDNA extracts were stored at −20 °C until further downstream analysis. The DNA for shotgun sequencing was extracted using DNeasy PowerMax Soil DNA and the DNeasy Power Soil DNA isolation kits (Qiagen) from 14 sediment samples distributed from top to bottom of the long core (Supplementary Table 2 and Supplementary Note 1). These extractions were performed at the paleogenetic laboratories at the Alfred Wegener Institute (AWI) in Potsdam. The extraction protocols were modified[66], briefly, prior to adding the sediment sample, solution C1, 400 μL of proteinase K (2 mg mL$^{-1}$), and 100 μL of Dithiothreitol (DTT) (5M) were added and vortexed for 10 min.

**Quantification of cyanobacteria abundance and analysis of cyanobacteria DNA preservation.** Total cyanobacteria were quantified with the nonselective fluorescent dye SYBR Green quantitative PCR (qPCR) assay, which specifically amplified the cyanobacteria 16S rRNA-ITS (internal transcribed spacer) region using the primers CSIF (5'-GYCACGCCCGAAGTCRTTAC-3') and 373R (5'-CT AACCACCTGAGCTAAT-3')[67,68]. Briefly, qPCR programmes were followed by a melting curve step from 70 °C to 95 °C at a transition rate of 1 °C per 5 s in order to determine the amplification specificities. The qPCR assays were performed in triplicates on a CFX96 real-time thermal cycler (Bio-Rad Laboratories Inc., USA). The 16S rRNA-ITS copy numbers were calculated in line with Savichtcheva et al.[69], and the obtained values were mean triplicates of each sample expressed as total cyanobacteria normalised to extracted DNA (copies ng$^{-1}$ DNA) since sedaDNA concentrations varied with sediment depth. The amplified products were confirmed and analysed for unspecific PCR amplification with agarose gel electrophoresis.

In order to assess whether DNA fragmentation occurred after deposition and whether this fragmentation could have negatively influenced PCR efficiency[70], we compared the cyanobacteria composition of the long (~1500 bp) amplicons with that of short to medium (~350 bp) amplicons on samples from sediment depths of 265 (CE 1140), 490 (2380 cal. a BP), and 1105 (10,820 cal. a BP) cm. PCR amplification of the long fragments was performed by targeting the 16S rRNA-ITS genomic region (PCR targeting of the entire cyanobacteria community) using the primers CYA371F (5'-CCTACGGGAGGCAGCATGTGGGGAATTTTCC-3') and 373R (5'-CTAACCACCTGAGC-3')[69]. The sequencing of short cyanobacteria fragments was performed on the amplification products from total cyanobacteria 16S rRNA-ITS qPCR assays. The clone libraries were constructed using a TOPO TA cloning kit (Invitrogen) with PCR vector 2.1 in line with the manufacturer's instructions. A total of 38 long fragments and 58 short fragments of cyanobacteria clones were sequenced for samples from sediment depths of 265, 490, and 1105 cm (Beckman Coulter Genomics, Hertfordshire, UK).

**Library preparation for amplicon sequencing.** The PCR for the Illumina amplicon sequence libraries was conducted using the cyanobacteria-specific primers CYA359F (5′-GGGGAATYTTCCGCAATGGG-3′) and CYA784R (5′-GAC-TACWGGGGTATCTAATCCC-3′)[21], which amplify a >400-nt-long fragment of the V3–V4 regions of the 16S rRNA gene. The primers had unique tags that served to differentiate the samples. The samples and negative controls (i.e., a reaction with PCR water as a template) were amplified in a 50-μL PCR reaction that consisted of 5x Platinum$^{TM}$ II PCR Buffer (Invitrogen, ThermoFischer Scientific), 25 mM of MgCl$_2$, 0.2 mM of deoxynucleoside triphosphate (dNTP) mix (ThermoFisher Scientific), 0.5 mM of each primer (TIB Molbiol, Berlin, Germany), and 2 U of Platinum$^{TM}$ II Taq Hot-Start DNA Polymerase (Invitrogen, ThermoFischer Scientific). The volume of the template DNA used in each reaction varied between 1 and 4 μL depending on the genomic DNA concentration. The PCR programme included a first denaturation step at 94 °C for 2 min followed by 35 cycles at 94 °C for 15 s, annealing at 60 °C for 15 s, and extension at 68 °C for 15 s. To avoid cross-contamination, older sedaDNA samples were amplified at different times than younger ones. Furthermore, to control for reproducibility of the PCR and sequencing results, all samples were amplified in a second PCR run (technical replicates). The tagged PCR products were then purified with the Agencourt AMPure XP kit (Beckman Coulter, Switzerland) and eluted in 30 μL of DNA/RNA-free water. The purified product was quantified with a Qubit 2.0 Fluorometer (ThermoFisher Scientific). Equimolar concentrations of all samples, their technical replicates, and their purified negative PCR controls were pooled into 2 separate multiplex libraries (à $n = 74$ samples and 2 negative controls). The

libraries were pair-end sequenced ($2 \times 300$ bp) on an Illumina MiSeq system at Eurofins Scientific (Constance, Germany).

**Amplicon sequence data processing**. Sequence raw reads were demultiplexed as well as adapter- and quality-trimmed via cutadapt v3.4 using the pair-end mode and the following parameters: -e 0.2 -q 15,15 -m 150–discard-untrimmed. The ASVs were generated using trimmed reads and the DADA2 package v1.20[71] using the pseudo-pooled approach with the following parameters: truncLen=c(240,200), maxN = 0, rm.phix = TRUE, compress=TRUE, multithread = TRUE, minLen = 150 with R v4.1. Taxonomic assignment was performed using DADA2 and SILVA database v138. At the genus level, there was 98% agreement between 50-s and 80-s minBoot and 95% agreement at the family level. The number of input, processed, and final reads of the DADA2 pipeline is shown in Supplementary Table 1. Subsequently, ASVs that represent chloroplasts, mitochondria, singletons, doubletons, other bacteria, rare taxa were removed. The sum of the sequencing reads from the two extraction blanks and 2 PCR negative controls (simply PCR blanks with only pure water) were subtracted from the samples before further downstream analyses. The 74 sedaDNA samples resulted in a total of 9,770,821 denoised, nonchimeric, and error-corrected sequences that DADA2[71] inferred in 9548 ASVs. In total, the filtered data set comprised 441,045 sequence reads in 433 ASVs that were assigned to photosynthetic cyanobacteria and distributed across 74 samples. Of the 433 ASVs, 9 were assigned to the order level, 45 to the family level, and 393 to the genus level (91% of all cyanobacteria ASVs; 436,981 read counts).

**Shotgun sequencing and bioinformatics**. The 14 sedaDNA samples for shotgun sequence analysis were sequenced at Eurofins Scientific (Constance, Germany) on an Illumina NovaSeq6000 machine that aimed for 35 million pair-end reads of 150 nt in length. The sequencing depth differed between samples due to input DNA concentration, with the deepest sediment samples having the lowest DNA concentrations (Supplementary Table 2). The raw reads were processed using the ATLAS metagenome pipeline[72] in order to obtain dereplicated, quality-controlled, and trimmed reads. These reads were mapped to the SILVA 16S SSU database v138[73] in order both to obtain taxonomy and to calculate species abundance. Mapping was performed using bowtie2 v2.4.2[74]. Cyanobacteria-affiliated nitrogen fixation (*nif*) gene family were analysed in the shotgun-sequences, to trace the long-term abundance of cyanobacteria-affiliated *nif* genes in Lake Tiefer See, and compare how these correspond to the abundance of diazotrophs (e.g., *Aphanizomenon*) revealed by amplicon sequencing. Microcystin biosynthesis (*mcy*) gene family presence was assessed to trace the long-term abundance of cyanobacterial toxin genes in the lake, and compare this to the abundance of known producers of the *mcy* toxin, e.g., *Planktothrix* and *Microcystis* revealed by amplicon sequencing. To obtain cyanobacteria-affiliated *nif* and *mcy* biosynthesis gene abundances, quality-controlled reads were assembled into contigs using the ATLAS metagenome pipeline[72]. The genes were predicted using prodigal v2.6.3[75] and annotated using the eggNog emapper v2.0.1[76] with a database from October 2020[77]. TaxonKit v0.8.0 was used to obtain lineage information from taxon IDs of annotated genes[78]. To obtain the final gene abundances, quality-controlled reads were mapped against all assembled contigs using bowtie2 v2.4.2[74]. The obtained counts were normalised as follows: First, gene counts were corrected for sequencing depth by normalising to the total number of sequencing reads of each sample. Second, to correct for different gene length, counts were normalised by kilobases of gene length. These normalised gene counts represent gene abundance that is comparable between different genes both within and between samples. To analyse shotgun ancient DNA damage we downloaded all Cyanobacteria genomes from NCBI and mapped the QC-controlled reads against all genomes using Bowtie[74]. MapDamage2[60] was executed on the mapped reads. The fragmentation pattern was extracted from the mapDamage2 output, then the mean and cumulative sum over the fragment sequence length was calculated.

**Statistics and reproducibility**. The details of the experimental design and statistical tests used in the study are described in the respective results and methods sections. The two amplicon sequence libraries were merged by taking the average of their relative abundances. Bubble plot was used to illustrate changes in cyanobacteria community composition since the beginning of the Holocene to present. The Bubble plot was produced with the open-source software Shiny (http://shiny.raccoome.de/bubblePlot/). Alpha- and beta-diversity estimations as well as multivariate permutational analyses (PerMANOVA) based on amplicon data were performed using the PAST v4.01 software[79]. Non-metric multidimensional scaling (NMDS) was performed using the "vegan" package in R[80]. Moreover, before performing beta-diversity estimations, the ASV cut-off was set to 0.1% in order to eliminate very rare taxa. Statistical analysis of shotgun sequence data was performed using R v3.6.2 with the clusterProfile 3.14.3[81] package, and the ggplot2 v3.3.2 package was used for visualisation.

To assess whether the cyanobacteria communities at the ASV level showed clustering patterns related to three major time periods in the last ca. 11,400 years, an NMDS analysis using Hellinger-transformed cyanobacteria absolute-read-count data as dependent variables was performed[82]. The time periods were defined based on changes in climate as revealed by continental paleoclimate data[46,52] and human-impact data in the study site derived from archaeological findings[37,38] and

vegetation openness reconstruction[27]. The three defined time periods were thus: (I) ca. 11,340–4070 cal. a BP, (II) ca. 3960–100 cal. a BP, and (III) ca. CE 1870–2000. The significance of the clusters was analysed with a non-parametric PerMANOVA based on Bray–Curtis using the time periods as predictors and Hellinger-transformed cyanobacteria absolute-read-count data as response variables[82]. A subsequent pairwise post-hoc Monte Carlo permutation test ($n = 9999$) was used to assess significant differences among the clusters. To assess the correlation of cyanobacteria richness and abundance to environmental parameters, a Spearman rank-based correlation coefficient was calculated. Prior to the correlation analysis, the environmental data (predictors) had been standardised by subtracting the mean and dividing by the standard deviation (Z-score), and cyanobacteria richness and abundance were used as response variables. Collinearity in the explanatory variables was tested with a variance inflation factor (VIF) using the "vif.cca" function in "vegan"[80]. Explanatory variables were then additively tested until only those with a VIF score < 10 remained (Supplementary Table 4). The significant subset of explanatory variables that may explain the variability of cyanobacteria community composition was determined via forward selection using the "ordiR2step" function in "vegan".

**Reporting summary**. Further information on research design is available in the Nature Portfolio Reporting Summary linked to this article.

## Data availability
The amplicon and shotgun sequencing raw reads have been uploaded to the European Nucleotide Archive (ENA) under BioProject accession number PRJEB51951. The filtered sedaDNA datasets analysed during this study are available in Supplementary Tables 1 and 2. The cyanobacteria lipid biomarker 7-methylheptadecane data analysed for this study are available in Supplementary Table 2. The age-depth model and pollen-based vegetation openness data of the last 6000 years for Lake Tiefer See are available at https://doi.org/10.1594/PANGAEA.862115.

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

## Acknowledgements
We thank the German Environmental Foundation (Deutsche Bundesstiftung Umwelt DBU) for funding the research position of E.C.N. We thank the coring team of GFZ-section Climate Dynamics and Landscape Evolution who assisted in the fieldwork (Brian Brademann). We thank Anke Saborowski for laboratory support with TOPO Cloning analysis.

## Author contributions
E.C.N., A. Bartholomäus, and S.L. designed the study. S.P. and E.C.N. conducted fieldwork. E.C.N. carried out molecular laboratory analyses. A. Bartholomäus and S.P. provided lake sedimentary data. A. Bartholomäus provided sediment age model data. M.T. provided reconstructed vegetation openness data. J.K. provided cyanobacteria lipid biomarker data. K.R.S. and T.W. performed DNA extraction for metagenome sequencing. J.-P.S. provided archaeological data. A. Brauer preprocessed the amplicon and metagenome sequence data. D.W. provided additional funding. E.C.N., A. Bartholomäus, M.-E.M., A. Brauer, and S.L., led the interpretation. E.C.N. wrote the first draft of the manuscript. All authors discussed the results and provided intellectual input to the manuscript.

## Funding

## Competing interests
All authors declare no competing interests.
