## [Peer Review File · Communications Biology]

Reviewers' comments:

Reviewer #1 (Remarks to the Author):

General comments

This paper uses the techniques of sediment DNA analyses to document the impacts of Bronze Age settlements on cyanobacterial abundance and composition. Specifically the authors extend the use of qPCR to quantify total cyanobacterial abundance developed in earlier studies such as Savichtcheva et al. (2011) and Pal et al. (2015), and the use of shotgun sequencing/HTS as in Monchamp et al. 2016 and Pilon et al. 2019 to analyze beyond the past ~ 200 years to a remarkable ~ 3800 y BP. The first 350 years in Lake Tiefer See (or simply Tiefer See? since lake and see are redundant terms?) were recently presented in Nwosu et al. (2021), also using sedDNA. The results of the present paper are clearly analyzed and presented for a succinct and sound story that should attract interest. The paper is well written (although some important points are missed and some statements were unclear (plus a bit vague) or unsupported – examples are given below).

However, there have been other papers documenting the impact of Bronze age settlements on cyanobacteria development based on the use of cellular remains. Although not sedDNA, these papers are worth referencing and discussing as they are thematically quite relevant. Hillbrand et al. (2014) (Non-pollen palynomorphs show human- and livestock-induced eutrophication of Lake Nussbaumersee (Thurgau, Switzerland) since Neolithic times. The Holocene 24: 559–568) showed detailed profiles of Anabaena, Aphanizomenon, and Gloeotrichia akinete concentrations where peaks correspond to Neolithic settlements 3915–3690 y BP and subsequent settlement periods. Interestingly they concluded on the resilience of the lake studied following abandonment of the Bronze Age settlements as opposed to the more permanent shift towards cyanobacteria described in the present paper. Another study of ancient settlement effects on cyanobacteria is also Gosling et al. (2019). So, the conclusion of the present paper is not novel per se but the techniques used, and the supporting evidence collectively provide a compelling story.

Specific comments

Line 95. The term vegetation openness will not be familiar to a lot of readers. Explain here.

Line 99-104. Why provide the findings of the paper and the take-home message in the Introduction here? especially when the leading sentence of the paragraph is about the objectives of the paper. I would delete lines 99-104 and use the space to provide the environmental context and history of the region.

Line 120 : “ lack of major inflow” so what is the source of water to this lake? From Fig. 1 it almost seems like it is part of a chain of lakes or was at one time? Explain if the hydrology of the system and/or climate have changed over the same time period (water level changes?)

Line 120. “detrital components are rare”? Detritus can mean simply decomposing organic matter or does it have a particular geological meaning here?? in which case please explain or use a different term for the mineral component of these sediments.

Line 126. Replace “parameters” with variables as these are not strictly speaking parameters (ask a mathematician). These terms have different meanings.

Lines 132. How is this reflected in the pollen composition? Somewhere how “openness” is quantified should be explained.

Line 145, normalized by wet weight not dry weight? Why is this? Water content would be much higher at the top than the bottom affecting these concentrations such they are not truly comparable.. Good to show total cyanobacteria as copies per unit DNA but this should be justified more explicitly either here or in the Materials and Methods (line 440).

Corrected for temporal autocorrelation?

Line 204. Please provide some justification for the choice of these genes

Line 217. These are picocyanobacteria and ubiquitous and more abundant than larger bloom forming taxa in lakes (often by several orders of magnitude).

Line 218. Not surprising Picocyanobacteria like Cyanobium are ubiquitous and always more abundant (by several orders of mag). This might help reader understand why it is so dominant throughout the

core (although in some other lake cores this is not always seen...)

Lines 232-234. Cyanobacteria are present in all waters. So this is hardly surprising?? Not sure I understand the point here. Delete second clause or reword.

Line 239. Well they are super ancient!

Lines 248-250. Why is this detail relevant? Collapse into the next sentence.

(line 256 seen in the pollen data??).

Line 265. This would not be the only source of nutrients. Nutrients from land clearing itself via erosional inputs or via use of fires for preparing land for agriculture (ash contains P)? post water level changes/flooding inputs?

Line 266. This was also the case back in the Bronze Age- how is this known for a fact?

Line 267. Why is diversity of interest here? other microbes can contain the nif genes so perhaps reword since you examined only cyano nif

Line 270. Would this then point to low/less nitrogen availability?? (not likely so other co-drivers of N₂ fixers?).

Line 276. If mass concentrations are reported here. why the % symbol

Fig. 5. Provide detection limits for estimates of gene copy numbers. Was myc gene family at this detection limit prior to 1900? Or were no samples analyzed? (similarly for the time period ~ 530-880 This could be better Indicated in Fig. 5 or at least in legend as an explanation. Fig. 5 is interesting as it suggests in modern times there has been a big jump in toxigenic taxa?

Line 272. What changing climate? I think you need to be a bit clearer here as to what you are referring to.

Line 286. Why is this? because no change in nutrient loading or stoichiometry of loading? So only climate change left as an explanation?

Line 301. Any evidence from this or other lakes in the region of increasing water temperatures? From monitoring record (if not of water temp then air temp which correlates with water temp, although actual relationship varies geographically).

Line 302-304. This segue in the form of supporting evidence is a bit tenuous. This (citation) is empirical correlative evidence based on contemporary monitoring across many lakes, so the verb tense here (in the past which makes it sound that this is based on a paleo type record) is a bit odd. So the scales (both temporal and spatial) are completely different, so perhaps keep to a general statement in the present tense and something more speculative as this is not a causal mechanism "...nutrients mainly control cyano ...in mesotrophic lakes, while temperature may be a more important driver in mesotrophic lakes..."

Line 304 correct spelling of phosphorus. Line 305-306 with all the studies done on Tiefer See it is surprising that this is not known with certainty.

Line 310. Use the plural Human impacts since then have...

Line 318...So what was the detection limit of these assays? and indicate in Fig. 5 that dates before 1900 were indeed tested and were at the detection limit....Reword line 319- this is an obvious statement..and speaks to either quantity or quality of DNA at these depths or to the fact that the gene was effectively not present.

Line 322 and the main contemporary producer in lakes of northern Germany...see Dolman et al. 2012. Which is not necessarily the case for other lake districts so I think this is worth mentioning.

Line 326 without the other biomarker data for cyanobacteria the multi-proxy data are not really presented?

Line 320 what time? The 1,900 mark?

Line 335. The term robust is a bit overworked in the text (see also line 356 etc...) please find a more specific term? Why/how is the signal "robust" and what does this mean here?

Line 339 -340- how was "varve quality" correlated with ASV richness? And what does this mean?

Line 350-352. This sentence is not clear. The fact this biomarker 7-methylheptadecane is found in other cyanos. Is this because there are peaks in the former that don't correspond to Aphanizomenon?

Line 355. What does "recent" mean here? my understanding is that you are considering the prior to the Bronze era or is this in reference to time before industrialization...

Line 377. Add lake depth at the coring sites - the depth of the site can have a bearing on DNA

preservation.

Line 410. In this section explain the resolution for the sediment sections used for DNA analyses. Provide the amount of sediments used for DNA extraction because it is quite remarkable that multiple sediment extractions were not required to be pooled to obtain sufficient (and good quality) DNA for analysis when the sediments were as old as the ones towards the bottom of the core. And at the bottom basal samples there were problems with the low number of reads? The latter point does not entirely jive with some of the text. It is critical to include sample sizes (while details of extraction, PCR methods etc.. can be simply provided through reference to previous studies using similar methods).

Line 420 what was the negative control here?

Additional comments:

Table 1 could go in Supp material it is not terribly informative/illustrative and it can be summarized in the text.

The biomarker 7-methylheptadecane reconstruction would be good to include in the paper itself!

Perhaps Fig. 2 can have a couple of panels or ditch the column for number of years per sample (and include/explain in text)?

Fig. 3 legend is pretty cursory and needs further explanation. Some of the dots are tiny (and not on the scale to the left). Are they representative of 10% or 1% or ~ 0.1% abundance? (explain in legend...)

Reviewer #2 (Remarks to the Author):

The paper "Early human impact on lake cyanobacteria revealed by a Holocene record of sedimentary ancient DNA" by Nwosu et al. uses sedimentary ancient DNA and other palaeoecological proxy methods to reconstruct the history of cyanobacteria throughout the Holocene in a German lake. They find evidence suggesting an increase in cyanobacteria associated with human activities as early as ~4000 years BP, implying that there has been a lengthy legacy of human impacts on ecosystems far exceeding the industrial revolution.

I congratulate the authors on such a thorough study. The writing, figures, and analyses are all of high quality. Their combination of methods and tight focus on cyanobacteria as it relates to archaeological human activities is novel and I recommend the article for publication.

I have just a few questions/requests that I think will help further clarify the work, which hopefully won't be too onerous. My comments are focused on the sedaDNA data as this is my area of expertise.

1) The main claim is that cyanobacteria increases in abundance after 4000 BP. This is largely based on qPCR with other associated supporting data that this measure is accurate. I think your lines of argumentation and complementary data are largely convincing that these qPCR-derived measures are reasonably specific to cyanobacteria, and that their DNA does increase in the lake over time at a rate exceeding the overall increasing preservation of sedaDNA nearing the present. Supp Fig 4. (the shotgun data) however makes it seem as if there isn't much change in cyanobacterial abundances because the graph is scaled by percentage. Figure 5 helps support the rise in specific genes (assuming this is the shotgun data, I think you should make the caption a bit clearer for this figure), but I'm curious what your abundances would look like based on read counts? When your libraries are normalized, do read counts of cyanobacteria increase through time relative to other bacterial families? Based on your qPCR, you should see this in your shotgun data as well I would think. I realize that deriving abundances from shotgun data is fraught with many interpretive limitations, but those same limitations are largely already in your qPCR data. Can you produce a companion figure to Supp Fig 4. that displays normalized read counts overall?

I also wonder to what degree smaller fragment sizes in your older material may have been lost during

your kit-based extractions, artificially biasing your qPCR data to having more and more cyanobacterial DNA (larger fragments) in younger time periods?

2) You didn't spend much time in the paper discussing why the cyanobacterial sedaDNA you're getting is actually ancient and not the result of some sort of contamination. Maybe I've missed it, but I didn't see a discussion of your blanks/negative controls. They don't seem to be included in any of your plots or tables, even though you say that you ran a blank for every 7 samples (L419-420). Judging by how many samples you have, you should have a lot of blanks by that ratio. Can you include an analysis of your negative control seq data to help demonstrate that these results aren't the result of contamination?

I'm surprised by how large of fragments you sequenced/targeted from the amplicon data, far exceeding what's typical of early-to-middle Holocene aDNA fragment length distributions. Perhaps it's because they're from bacteria and they have sustained repair mechanisms that have stopped the fragments from getting too small. You sequenced quite deeply in the 14 shotgun samples. Deeply enough that I assume you could map your data to larger portions of cyanobacteria genomes than just 16S. Is it possible to do so in order to generate 1) mapDamage misincorporation plots to investigate deamination in your data (supporting that these sequences are ancient), and 2) fragment length distributions to look at rates of depurination?

It would also be interesting (although not necessary if you think it's beyond the scope here) to use the approach from Kistler et al. 2017 (<https://doi.org/10.1093/nar/gkx361>) to look at how damage patterns from that shotgun data change through time relative to age/temperature. This data would help support that these cyanobacteria sequences are ancient, and ideally, that the amount of damage is roughly correlated with age, temperature, or other local factors influencing rates of aDNA damage. This would also help evaluate to what degree the shifting diversity you observe in cyanobacteria genera is more so related to changing intensities of human impacts or simply that the DNA isn't as degraded because it's younger (and hence you find more of it). These FLDs and/or depurination analyses will help with my previous question as to whether you may have lost more of your ancient DNA in the older samples, which could have artificially contributed towards the apparent increase in cyanobacteria DNA in the younger material.

It could also be interesting to run some phylogenetics with your mapped cyanobacteria shotgun data. This again might be beyond the scope here, but could be quite interesting with the data already available if you have the mapped coverages.

I realize this paper is focused on cyanobacteria, but I'm also curious what eukaryotic DNA was observed in the shotgun data? Can that be used at all to help support the claims that these shifts are driven by human activity (agricultural plants/animals showing up, human DNA, decrease in taxa susceptible to cyanobacterial toxins [like fish], etc.). This may be for another paper, but if not, it would strengthen your claims that these shifts are related to human activities and not climate. There are presumably many other organisms contributing to the changing ecology of this lake.

L147. The word concentration used twice, two words apart, consider changing to something like "The main finding is a distinctly higher sedaDNA concentration since..."

Fig 4. The caption doesn't make clear that this is from the shotgun data. It would also be nice if you included depths like in figure 3 so that the samples can be easily correlated with your supplementary tables.

Sup Fig 1. It would be helpful to make the labels a bit more readable on this figure, similar to the main text map (which is beautiful by the way).

Great job! It was a pleasure to review this paper. Your manuscript has helped give me ideas on how to

best analyze microbial sedaDNA data that I'm also working with.

-Tyler Murchie

Referee expertise:

Referee #1: lake palaeology (among other things)

Referee #2: sedaDNA (among other things)

Reviewers' comments:

Reviewer #1 (Remarks to the Author):

General comments

This paper uses the techniques of sediment DNA analyses to document the impacts of Bronze Age settlements on cyanobacterial abundance and composition. Specifically the authors extend the use of qPCR to quantify total cyanobacterial abundance developed in earlier studies such as Savichtcheva et al. (2011) and Pal et al. (2015), and the use of shotgun sequencing/HTS as in Monchamp et al. 2016 and Pilon et al. 2019 to analyze beyond the past ~ 200 years to a remarkable ~ 3800 y BP. The first 350 years in Lake Tiefer See (or simply Tiefer See? since lake and see are redundant terms?) were recently presented in Nwosu et al. (2021), also using sedDNA. The results of the present paper are clearly analyzed and presented for a succinct and sound story that should attract interest. The paper is well written (although some important points are missed and some statements were unclear (plus a bit vague) or unsupported – examples are given below).

Thank you for taking the time to read our manuscript and providing suggestions, comments and questions that helped us to improve it. In publications so far, the lake has been referred to Lake Tiefer See, and although we agree that there is some redundancy in the name, we would like to keep it that way to maintain uniformity with previous publications.

However, there have been other papers documenting the impact of Bronze age settlements on cyanobacteria development based on the use of cellular remains. Although not sedDNA, these papers are worth referencing and discussing as they are thematically quite relevant. Hillbrand et al. (2014) (Non-pollen palynomorphs show human- and livestock-induced eutrophication of Lake Nussbaumersee (Thurgau, Switzerland) since Neolithic times. *The Holocene* 24: 559–568) showed detailed profiles of Anabaena, Aphanizomenon, and Gloeotrichia akinete concentrations where peaks correspond to Neolithic settlements 3915–3690 y BP and subsequent settlement periods. Interestingly they concluded on the resilience of the lake studied following abandonment of the Bronze Age settlements as opposed to the more permanent shift towards cyanobacteria described in the present paper. Another study of ancient settlement effects on cyanobacteria is also Gosling et al. (2019). So, the conclusion of the present paper is not novel per se but the techniques used, and the supporting evidence collectively provide a compelling story.

We appreciate the suggested studies and agree that they are thematically relevant. We have referenced the studies by Hillbrand et al. and Gosling et al. in the introduction on line 56 “Early cyanobacteria paleoecological studies partly dating back to ~10,500 years ago commonly used fossilized remains (akinetes, cysts, Hillbrand, Gosling, and other previously referenced studies), pigments, and biomarkers, preserved in sediments as proxies for inferring past changes.”

In the discussion, we addressed the reviewer’s comment and added text to discuss previous studies, which can be found on lines 296-333 in the Discussion section.

Specific comments

Line 95. The term vegetation openness will not be familiar to a lot of readers. Explain here.

We have now clarified the meaning of vegetation openness by adding “land cover changes inferred from pollen data (Theuerkauf et al 2016)” on lines 95-96.

Line 99-104. Why provide the findings of the paper and the take-home message in the Introduction here? especially when the leading sentence of the paragraph is about the objectives of the paper. I would delete lines 99-104 and use the space to provide the environmental context and history of the region.

These lines have been removed and replaced with environmental context and history of the region. The following sentence were added on lines 100-105 “Within the Lake Tiefer See catchment and wider region settlements by sedentary farmers and ranchers appeared during the Bronze Age ca. 3940–3100 cal. a BP (Schmidt 2017 and Raese und Schmidt 2019). During the Iron Age the lake area was largely depopulated, this period is then followed by intermittent human occupation during ca. 2700–2200, 990–895 (CE 960–1055), and since 750 cal. a BP (CE 1200), with settlements preferentially built on the banks of water.”

Line 120 : “lack of major inflow” so what is the source of water to this lake? From Fig. 1 it almost seems like it is part of a chain of lakes or was at one time? Explain if the hydrology of the system and/or climate have changed over the same time period (water level changes?)

The following sentences were added to the Methods section under the subheading Lake hydrology on lines 512-520.

“The source of water of Lake Tiefer See is mainly rainfall and to some extent subsurface flow and groundwater. Lake Tiefer See is part of a chain of lakes but the connection to the northern Lake Flachersee was piped during railway constructions from CE 1884–1886 and is in modern times mostly dry (Kienel et al. 2013). The present connection to Lake Hofsee in the South is through a wetland and is very shallow (Fig. 1). According to the reconstructed water level changes throughout the Holocene, lake levels were always at least 1–2 m below the range of present-day fluctuations until ca 2,800 cal. a BP so that Lake Tiefer See could not have been connected to Lake Hofsee during the Bronze Age.”

Line 120. “detrital components are rare”? Detritus can mean simply decomposing organic matter or does it have a particular geological meaning here?? in which case please explain or use a different term for the mineral component of these sediments.

What we refer to is the transport of detrital minerogenic sediments (clay to silt-sized) into the lake which is very low due to the lack of major inflows (Roeser et al 2021). Accordingly, we have reworded to “detrital minerogenic components are rare” on line 126.

Line 126. Replace “parameters” with variables as these are not strictly speaking parameters (ask a mathematician). These terms have different meanings.

“Parameters” is now replaced with “variables” on line 132.

Lines 132. How is this reflected in the pollen composition? Somewhere how “openness” is quantified should be explained.

We derive changes in vegetation openness from changing proportion of pollen from herbal taxa, such as pollen from wild grasses, cereals, *Plantago lanceolata* or *Rumex acetosa/acetosella*. However, because the original pollen percentages do not 1:1 reflect past vegetation composition and particularly underestimate the proportion of past open vegetation, we instead use the REVEALS approach here for correcting raw pollen data for this effect. The underlying methodology has now been added on lines 543-547, “The past vegetation openness was reconstructed from pollen percentage data using the REVEALS model (Sugita, 2007). The model was applied with the REVEALS in R function from the DISCOVER package with the Lagrangian stochastic model for pollen dispersal and the PPE.MV2015 data set (Theuerkauf et al., 2016)”

To avoid confusion about the term openness, we have rephrased the sentence to:

“The pollen-based vegetation reconstruction shows repeated changes in the proportion of forested versus non-forested, that is open vegetation such as grasslands or arable land, in the pollen source area of Lake Tiefer See (Fig. 2) on lines 137-139”.

Line 145, normalized by wet weight not dry weight? Why is this? Water content would be much higher at the top than the bottom affecting these concentrations such they are not truly comparable.. Good to show total cyanobacteria as copies per unit DNA but this should be justified more explicitly either here or in the Materials and Methods (line 440).

Corrected for temporal autocorrelation?

We sub-sampled the long core immediately after opening, while the sediment samples used to estimate water content were taken later, during which time the cores have dried a bit and thus have lower water content than in the samples we processed for DNA analyses (amplicon and qPCR). This is the reason why we normalized our sedaDNA content to wet weight, like we did in an earlier study (<https://doi.org/10.3390/microorganisms9081778>). However, we agree that the water content would probably be higher in the top sediment layers compared to the deeper layers. But to demonstrate that copy numbers per dry weight would be even higher in the younger sediment compared to the older and deeper sediment, we show below a plot of sedaDNA per wet and dry sediment (please note dry weight could not be assessed for all samples) as well as cyanobacteria copies normalized to wet and dry sediment weight. As suspected, copy numbers were higher when normalized to dry weight. Regarding the normalization of gene copy numbers to unit DNA in the present manuscript, we have added the following explanation in the manuscript on line 602 “...since DNA concentrations varied with sediment depth”.

Thus, we decided against using normalized cyanobacteria copy numbers to dry weight in the manuscript because the water content was estimated from sediments with lower water content to the samples, we processed for DNA analyses.

Line 204. Please provide some justification for the choice of these genes

In accessing the presence of cyanobacteria-affiliated *nif* genes in the shotgun sequenced samples, we wanted to trace the long-term abundance of the nitrogen-fixing genes in Tiefer See and see how these correspond to the amplicon abundance of diazotrophic cyanobacteria (e.g. *Aphanizomenon*). In assessing *mcy* gene presence, we wanted to trace the long-term abundance of cyanobacterial toxin genes in the lake, and compare this to the amplicon abundance of known producers of the cyanotoxin microcystin (e.g., *Planktothrix* and *Microcystis*). We have added this information to justify our choice of these genes at lines 674-681 of the revised manuscript. “Cyanobacteria-affiliated nitrogen fixation (*nif*) gene family were analyzed in the shotgun-sequences, to trace the long-term abundance of the *nif* genes in Lake Tiefer See, and compare how these correspond to the abundance of diazotrophs (e.g. *Aphanizomenon*) revealed by amplicon sequencing. Microcystin biosynthesis (*mcy*) gene family presence was assessed to trace the long-term abundance of cyanobacterial toxin genes in the lake, and compare this to the abundance of known producers of the *mcy* toxin e.g., *Planktothrix* and *Microcystis* revealed by amplicon sequencing.”

Line 217. These are picocyanobacteria and ubiquitous and more abundant than larger bloom forming taxa in lakes (often by several orders of magnitude).

We agree that the Picocyanobacteria are more abundant, but with this sentence we wanted to show result similarities between the two different sedaDNA approaches: shotgun and amplicon sequencing. Since this apparently was not made clear enough, we have modified the sentence following the reviewer’s suggestion on lines 230-233 to “For example, the family Cyanobiaceae (which contains the ubiquitous genus *Cyanobium*) was the most abundant in all the shotgun-analyzed samples similar to the amplicon sequencing for the same samples, in which *Cyanobium* was also most abundant.” And in the Discussion on lines 257-259 “The Picocyanobacteria (*Synechococcus* and *Cyanobium*) are generally more abundant (often by several orders of magnitude) than larger bloom-forming taxa in lakes (Callieri and Stockner, 2000).”

Line 218. Not surprising Picocyanobacteria like Cyanobium are ubiquitous and always more abundant (by several orders of mag). This might help reader understand why it is so dominant throughout the core (although in some other lake cores this is not always seen...)

We agree and have modified our sentence as presented in our answer to the previous comment.

Lines 232-234. Cyanobacteria are present in all waters. So this is hardly surprising?? Not sure I understand the point here. Delete second clause or reword.

We agree cyanobacteria are present in all lakes, and have now reworded the sentence on lines 253-254 to “Our sedaDNA reconstruction provides the first record of cyanobacteria DNA for the entire Holocene from a freshwater lake spanning the last ca. 11,340 years”.

Line 239. Well they are super ancient!

Indeed, they are. We think that your comment addresses the term “pioneer organisms” However, we use this term as a description of early colonizers of initial/newly formed ecosystems and not as a description for evolution. As a result, we have added an explanatory sentence on lines 266-267 “meaning that they are among the initial colonizers of newly formed ecosystems.”

Lines 248-250. Why is this detail relevant? Collapse into the next sentence.

This detail is relevant because it further demonstrates how exactly Bronze Age humans likely lived in the lake region, based on the archaeological findings close to the lake. It also shows a likely intensive impact on the lake itself, since these activities require water. Therefore, we would like to keep the sentence. However, we added brief context why this is relevant in lines 278-279 “...thus, their activity and life-style likely had a direct impact on the lake ecosystems.”

(line 256 seen in the pollen data??).

During the Bronze Age, cereal pollen is still very rare in the Tiefer See pollen record. The few pollen grains found do not allow inferences about cereal cultivation in the vicinity of the lake. The study in reference 38 instead relies on macrofossil data. We have added the word “likely” to this sentence on line 285.

Line 265. This would not be the only source of nutrients. Nutrients from land clearing itself via erosional inputs or via use of fires for preparing land for agriculture (ash contains P)? post water level changes/flooding inputs?

We agree with the reviewer and thank for the suggestion. We now included the other possible nutrient sources in the sentence on lines 327-331 and reworded the sentence to “The sources of excessive nutrient addition into Lake Tiefer See were likely from animal and human excrement, land clearing itself, and/or use of fires for preparing land for agriculture (ash contains phosphorus). These nutrients were accumulated in the lake after inundation due to the lack of a major outflow and likely stored in the sediments.”

We do not include erosional inputs, because we only have evidence for a slight increase in detrital components (still very little) which might be even only reworked/resuspended. We do

not include post-water level changes as we do not see any influence of water level changes on cyanobacteria and from our microfacies analysis, we can exclude flooding events.

Line 266. This was also the case back in the Bronze Age- how is this known for a fact?

We are sorry but unfortunately, we do not understand this comment. But if you were referring to the “lack of major outflow”, the lake level reconstruction by Theuerkauf et al 2021 shows a 1-2 m lower water level during the Bronze Age which is a strong indication that the lake was not connected to Hofsee, thus it has no outflow.

Line 267. Why is diversity of interest here? other microbes can contain the *nif* genes so perhaps reword since you examined only cyano *nif*

The sentence states “In addition to cyanobacteria abundance, cyanobacteria beta-diversity also changed...”, so, we already specified a reference to cyanobacteria diversity, and not general bacterial diversity. We have now modified and split into two sentences on lines 334-337 to “In addition to cyanobacteria abundance, cyanobacteria beta-diversity also changed substantially beginning with the Bronze Age as shown by NMDS (Fig. 4). Additionally, *nif* genes affiliated with cyanobacteria were detected for the first time at ca. 3,520 cal. a BP, as these genes were not found in earlier periods (Fig. 5).” Also, in the methods we added on lines 680-681 “...cyanobacteria *nif* and *mcy* biosynthesis gene...”, to make it clear that only cyanobacteria genes were examined.

Line 270. Would this then point to low/less nitrogen availability?? (not likely so other co-drivers of N₂ fixers?).

Unfortunately, we cannot make a definite statement on nitrogen availability (as we do not have this data), and this statement is based on the *nif* genes affiliated only with cyanobacteria. To make this clear we have added the following sentence on lines 338-342 “We consider the detectable increase in *nif* gene affiliated with cyanobacteria indicative of their increase in demand for nitrogen likely as a result of an increasing population. Either growth of cyanobacteria populations resulted in an increasing demand for nitrogen or the abundance of *nif* genes simply increased along with the increase in population size.” We agree that a bacteria kingdom-wide analysis may reveal *nif* gene expressions that resulted (or result) from other bacteria that are nitrogen fixers.

Line 276. If mass concentrations are reported here. why the % symbol

This was an error, which has been corrected in the revised MS on line 346 to 6.8 µg g TOC⁻¹.

Fig. 5. Provide detection limits for estimates of gene copy numbers.

Figure 5 shows data from shotgun sequencing and not from absolute quantification of gene copy numbers through, for example, qPCR. Therefore, the data refer to relative abundances. Here, the limit of detection depends on various factors, including library preparation efficiency and sequencing depth, which also influence data processing. Thus, a conclusive statement on the detection limit can, unfortunately, not be given, and is also not common for shotgun sequencing data.

A factor we can judge is the sequencing depth. This means that there could be a potential bias from low sequencing depth which coincides with the absence of *nif* gene counts. However, in

our study the relative abundance of *nif* gene counts was sometimes higher in samples with comparably low sequencing depth as shown in the table below.

Sediment composite depth (cm)	Age (cal. a BP)	Sequencing depth (reads in million)	% Gene counts/abundance of Cyanobacterial genes normalized to all Cyanobacterial genes
265	810	35	0,000
430	1,900	49	0,0118
531	2,660	29	0,0073
604	3,520	22	0,0135

The sample from 810 cal. a BP has a good sequencing depth (35mio reads) but no *nif* gene was detected. The sample from 3,520 cal. a BP has the lowest sequencing depth (22Mil. reads) but had more *nif* genes than the samples from 2,660 cal. a BP (29Mil. reads) and 1,900 cal. a BP (49Mil. reads). Therefore, our results suggest that sequencing depth is not the main factor. Please, also note that Fig 5. show normalized read counts of *nif* (and *mcy*) genes as described in the figure legend, thus sequencing depth effect is practically removed. For clarity, we have removed the apparent black lines in samples where no *nif* (or *mcy*) gene were detected as shown below. We have also added the following sentence to the figure legend on lines 1090-1091 “in samples with no bars, no cyanobacteria-affiliated *nif* genes (or *mcy*) were detected.”

Was myc gene family at this detection limit prior to 1900? Or were no samples analyzed? (similarly for the time period ~ 530-880 This could be better Indicated in Fig. 5 or at least in legend as an explanation.

For the answer to this question, please also refer to the statement above. All samples were analyzed for the presence of the *mcy* gene, but we could not detect the gene in samples prior to 1,900 cal. a BP, 1,280 cal. a BP, ~530–880 cal. a BP, and CE 1970. This explanation is now provided in the revised figure 5 legend.

Fig. 5 is interesting as it suggests in modern times there has been a big jump in toxigenic taxa?

Thank you for pointing out this important detail. Indeed, that is what Figure 5 suggests. We have stated this now more clearly on lines 401-402 "...and a substantial increase in toxigenic taxa since CE 1940 (e.g., *Microcystis*; Figs 3 and 5) ..."

Line 272. What changing climate? I think you need to be a bit clearer here as to what you are referring to.

The period ca. 2,610 cal. a BP referred to in this sentence corresponds to a period of increasing air temperatures after the Homeric solar minimum in the northern hemisphere (> 30°N, Büntgen et al., 2011, 2016). As a result, the sentence has been modified to "perhaps due to increasing air temperature after the Homeric solar minimum (Büntgen et al., 2011, 2016), or another unknown factor" on lines 344-345.

Line 286. Why is this? because no change in nutrient loading or stoichiometry of loading? So only climate change left as an explanation?

No, not just climate change is the explanation and anthropogenic eutrophication via agriculture (that peaked with pig farming in the 1970s) with its attendant nutrient loading. This bit of information is now added on lines 361-362 with references (Kienel et al., 2017 and Nwosu et al., 2021)

Line 301. Any evidence from this or other lakes in the region of increasing water temperatures? From monitoring record (if not of water temp then air temp which correlates with water temp, although actual relationship varies geographically).

Yes, there is evidence of increasing air temperature for the Lake Tiefer See region from the German weather station (Deutsche Wetter Dienst Station Schwerin for the period CE 1881–2017; 80km to the west of Lake Tiefer See). The evidence shows that the region is getting warmer (Page 15 Klimareport_M-V_20180802.indd (dwd.de)). In the revised manuscript on lines 377-378 we have added this information as follows "With climate warming in recent decades (meteorological data from DWD Station Schwerin CE 1890–2016; 80km to the west of Lake Tiefer See), increasing water temperatures may have played an increasing role."

Line 302-304. This segue in the form of supporting evidence is a bit tenuous. This (citation) is empirical correlative evidence based on contemporary monitoring across many lakes, so the verb tense here (in the past which makes it sound that this is based on a paleo type record) is a bit odd. So the scales (both temporal and spatial) are completely different, so perhaps keep to a general statement in the present tense and something more speculative as this is not a causal mechanism "...nutrients mainly control cyano ...in mesotrophic lakes, while temperature may be a more important driver in mesotrophic lakes..."

Thank you for the suggestion, we have now modified the sentence to be a more general statement on lines 379-381 "Our findings are supported by reports that nutrients mainly control cyanobacteria abundance in oligotrophic lakes while temperature may be a more important driver in mesotrophic lakes."

Line 304 correct spelling of phosphorus.

Phosphorus spelling was corrected on line 384.

Line 305-306 with all the studies done on Tiefer See it is surprising that this is not known with certainty.

This study focused on cyanobacteria DNA on Tiefer See since the early Holocene, is the first in a line of other paleo-reconstruction studies (e.g. *Chla*, for which fresh core collection is planned, and is to be used as an indicator of phytoplankton biomass production) planned for the same period to be conducted in the lake. The future studies when completed will inform with more certainty on the trophic state of Tiefer See during the early to mid-Holocene. Therefore, in the absence of these data, we can only assume that the lake was oligotrophic, especially as it had no major detrital mineralogenic input as mentioned above.

Line 310. Use the plural Human impacts since then have...

Human impacts now used on line 389.

Line 318...So what was the detection limit of these assays? and indicate in Fig. 5 that dates before 1900 were indeed tested and were at the detection limit....Reword line 319- this is an obvious statement..and speaks to either quantity or quality of DNA at these depths or to the fact that the gene was effectively not present.

We have indicated in the Legend of figure 5 that dates before ca. 1,900 cal. a BP were indeed tested. Our answer on why we cannot give a detection limit to the shotgun sequencing analysis is detailed above in our answer to your first question about figure 5. Thank you for the suggestion, we have now reworded to “The detection of *mcy* genes was not possible before likely because the gene was not present, or it was too low to be detected using a shotgun sequencing approach or too degraded due to degradation of ancient DNA over time. However, the high abundance of *mcy* genes in modern times, suggests a substantial increase in toxigenic cyanobacteria taxa in Lake Tiefer See” on lines 406-408.

Line 322 and the main contemporary producer in lakes of northern Germany...see Dolman et al. 2012. Which is not necessarily the case for other lake districts so I think this is worth mentioning.

Thank you for the suggestion. We have now added your suggestion and the reference on lines 401-402 “...and even the main contemporary producer in lakes of northern Germany.”

Line 326 without the other biomarker data for cyanobacteria the multi-proxy data are not really presented?

Originally, we meant the application of several DNA approaches, that is, qPCR, metabarcoding, and shotgun sequencing. We have now added the biomarker figure (updated

Fig. 2) to the main manuscript to justify our use of the term “multi-proxy” in this sentence.

Line 320 what time? The 1,900 mark?

Yes, around the ca. 1,900 cal. a BP Mark. We have now added this bit of information to the sentence on line 400.

Line 335. The term robust is a bit overworked in the text (see also line 356 etc...) please find a more specific term? Why/how is the signal “robust” and what does this mean here?

Originally, we used the term “robust” to emphasize the consistent increase in cyanobacteria gene copies in the samples between ca. 3,960 and 2,660 cal. a BP despite similar sedaDNA concentrations. We have now replaced “robust” with “reliable” on line 420, we hope this is a more specific term.

Line 339 -340- how was “varve quality” correlated with ASV richness? And what does this mean?

We used a Spearman correlations analysis as stated in the results section on lines 200-210 and also described in the methods section on lines 719-723. What this means is that varved sediments had more richness, that is, a higher number of different cyanobacteria taxa. This points to the fact that the formation of varves supports the DNA preservation as shown in Domaizon et al 2017, now stated in the revised manuscript on lines 424-426.

Line 350-352. This sentence is not clear. The fact this biomarker 7-methylheptadecane is found in other cyanos. Is this because there are peaks in the former that don’t correspond to *Aphanizomenon*?

Yes, you are correct, that is precisely what we meant. We have modified the sentence now on lines 437-439 to “other peaks in 7-methylheptadecane that do not correspond to *Aphanizomenon* may have been produced by other cyanobacteria producers of 7-methylheptadecane.” We hope this new formulation is clearer.

Line 355. What does “recent” mean here? my understanding is that you are considering the prior to the Bronze era or is this in reference to time before industrialization...

Yes, the latter point is what we meant, that is, a reference to the time before industrialization. We have added this bit of information in the revised manuscript on line 492 “...before industrialization...”

Line 377. Add lake depth at the coring sites – the depth of the site can have a bearing on DNA preservation.

The lake depth at the coring site, which is the deepest point in the lake (62 m) has now been added on line 525.

Line 410. In this section explain the resolution for the sediment sections used for DNA analyses. Provide the amount of sediments used for DNA extraction because it is quite remarkable that multiple sediment extractions were not required to be pooled to obtain sufficient (and good quality) DNA for analysis when the sediments were as old as the ones towards the bottom of the core. And at the bottom basal samples there were problems with the low number of reads? The latter point does not entirely jive with some of the text. It is critical to include sample sizes (while details of extraction, PCR methods etc.. can be simply provided through reference to previous studies using similar methods).

Thank you for this point which enabled us to present our sedimentary DNA extraction analysis more clearly. On lines 561-567 we have added the following “A total of seventy-four sediment depths were sampled from the middle of the cores. To avoid contamination, the top layers were scraped off after the core opening with a sterile razor and the uncontaminated anoxic part below was put into sterile 15-ml falcon tubes using sterile spatulas and stored at -80 °C until analysis. Cross-contamination of the sediments was avoided by processing recent and older samples at different times. Four total DNA extractions used for the qPCR assay and amplicon sequencing were performed per sample. The weight of the sediment that was obtained and used for each extraction was ~250 mg. All four DNA extracts per sample were pooled in a 2 ml Eppendorf Tube after extraction. The total DNA extraction was done...”

Line 420 what was the negative control here?

The negative controls were extraction blanks without sediment. This information has been added on line 578.

Additional comments:

Table 1 could go in Supp material it is not terribly informative/illustrative and it can be summarized in the text.

Thank you for the suggestion. Table 1 has been moved to supplementary material as supplementary table 5. A brief summary is now added in lines 198-200.

The biomarker 7-methylheptadecane reconstruction would be good to include in the paper itself! Perhaps Fig. 2 can have a couple of panels or ditch the column for number of years per sample (and include/explain in text)?

Thank you for the suggestion, we have added the biomarker data as part of Fig. 2 as shown above and removed the number of years per sample which is already part of Supplementary Fig. 2.

Fig. 3 legend is pretty cursory and needs further explanation. Some of the dots are tiny (and not on the scale to the left). Are they representative of 10% or 1% or ~ 0.1% abundance? (explain in legend...)

Thank you for pointing this out, the tiny dots represent ~0.1% abundance and below as now indicated in the new figure legend on line 1078. The dots representative of 10% and 1% are now also indicated on the scale to the left as shown below.

Reviewer #2 (Remarks to the Author):

The paper “Early human impact on lake cyanobacteria revealed by a Holocene record of sedimentary ancient DNA” by Nwosu et al. uses sedimentary ancient DNA and other palaeoecological proxy methods to reconstruct the history of cyanobacteria throughout the Holocene in a German lake. They find evidence suggesting an increase in cyanobacteria associated with human activities as early as ~4000 years BP, implying that there has been a lengthy legacy of human impacts on ecosystems far exceeding the industrial revolution.

I congratulate the authors on such a thorough study. The writing, figures, and analyses are all of high quality. Their combination of methods and tight focus on cyanobacteria as it relates to archaeological human activities is novel and I recommend the article for publication.

Thank you for your comments and for taking the time to read our manuscript. Your suggestions have helped us improve our manuscript by providing more clarifications where necessary and additional pictorial information in the supplementary material. More details are given to specific questions and comments below.

I have just a few questions/requests that I think will help further clarify the work, which hopefully won't be too onerous. My comments are focused on the sedaDNA data as this is my area of expertise.

The main claim is that cyanobacteria increases in abundance after 4000 BP. This is largely based on qPCR with other associated supporting data that this measure is accurate. I think your lines of argumentation and complementary data are largely convincing that these qPCR-derived measures are reasonably specific to cyanobacteria, and that their DNA does increase in the lake over time at a rate exceeding the overall increasing preservation of sedaDNA nearing the present. Sup Fig 4. (the shotgun data) however makes it seem as if there isn't much change in cyanobacterial abundances because the graph is scaled by percentage. Figure 5 helps support the rise in specific genes (assuming this is the shotgun data, I think you should make the caption a bit clearer for this figure) but I'm curious what your abundances would look like based on read counts? When your libraries are normalized, do read counts of cyanobacteria increase through time relative to other bacterial families? Based on your qPCR, you should see this in your shotgun data as well I would think. I realize that deriving abundances from shotgun data is fraught with many interpretive limitations, but those same limitations are largely already in your qPCR data. Can you produce a companion figure to Supp Fig 4. that displays normalized read counts overall?

Thank you for the question. First of all, we have made the source data of this figure clearer by adding “based on shotgun data” to the figure caption on line 246. Per your suggestion, we show below (part a) and in the revised supplementary data the normalized read counts overall of cyanobacteria changes relative to other bacteria and archaea. The shotgun data unfortunately does not provide a good temporal resolution as there are too few samples and the period showing the increase in cyanobacteria abundance in the qPCR data is not sufficiently resolved. Therefore, the trend seen in the qPCR data is not clearly visible in the shotgun data. However, the shotgun data confirm that the very early samples have hardly any cyanobacteria reads and that a substantial pool of cyanobacteria has developed until ca. 1,900 cal. a BP, consistent with the first Holocene peak in cyanobacteria abundance (qPCR) at ca. 1,900 cal. a BP (2.1×10^5 copies ng⁻¹ DNA) now shown in the revised manuscript on lines

478-482. The cyanobacteria reads in the samples corresponding to ca. 3,520 and 2,660 cal. a BP increased only slightly and as such do not match the trend shown by the qPCR data for the same period which we cannot fully explain except for the low temporal resolution as mentioned. So overall, the shotgun data support the qPCR data, although with limitations. Indeed, we initially decided against showing the cyanobacteria reads development relative to other bacteria and archaea because of the limitations in interpreting those data as you also mentioned, e.g., the likely presence of other active and recent bacterial and archaeal groups. These recent and/or active bacteria and archaea can be of varying abundances in different depths of the sediment, depending on the micro-habit conditions. Cyanobacteria can only be ancient, as they are known to need sunlight for photosynthesis, thus they cannot be active in sediments. Therefore, it is not surprising that cyanobacteria reads pale in comparison to other bacteria and archaea groups, especially in younger sediments. Fast-growing bacteria in these sediment layers that for example use nucleic acids, including those of lysed cyanobacteria cells, as a substrate would be represented by higher reads than cyanobacteria.

I also wonder to what degree smaller fragment sizes in your older material may have been lost during your kit-based extractions, artificially biasing your qPCR data to having more and more cyanobacterial DNA (larger fragments) in younger time periods?

Thank you for bringing up this important concern which enables us to provide further support on the signal of increasing cyanobacterial population sizes following human activity during the Bronze Age.

Following DNA extraction, we checked the quantity and quality of all DNA extracts through gel electrophoresis, including high-resolution chromatography-based Tape Station assays for deeper and older samples. Accordingly, the size range of recovered DNA between the youngest and oldest samples is comparable (from below 100 bp to highly genomic DNA), which is within the range (~100–500 bp) recommended by Boere et al., (2011)

<https://www.sciencedirect.com/science/article/pii/S0146638011002269?via%3Dihub> as appropriate to be analyzed from Holocene sediments. Therefore, sequencing and qPCR analyses were possible throughout the entire core independent of sample age. Shown below is the tape station result for 3 analyzed depths; confirming sedaDNA had sufficient fragment length for downstream processes which is now mentioned in the revised manuscript on lines 579-582.

The lower 100 bp signal in the tape station images below is from the marker, sample mean peaks are shown afterward i.e., 540, 660, and 579 for 3,690-, 5,070-, and 5,260 cal. a BP, respectively.

3,690 cal. a BP

5,070 cal. a BP

5,260 cal. a BP

We agree, however, that with increasing sample age there is increasing decay of aDNA. This is particularly visible based on DNA concentrations of the youngest samples spanning the time period III which are substantially higher than those of time periods I and II (Fig. 2). This is why we normalized our qPCR data against DNA concentrations as stated in the manuscript. DNA concentrations of time periods II and III, however, did not differ significantly, so the increase in qPCR gene copy numbers for cyanobacteria with the onset of time period II provides strong evidence that cyanobacterial populations relative to total measurable DNA increased as stated in the manuscript. Thus, if increasing fragmentation with increasing sample age was the reason for less amplification and thus a lack of detection through qPCR, we would have to expect a gradual decrease in gene copy numbers along the core. This, however, is not the case. Rather, we observe no change in qPCR data for the early to mid-Holocene periods but a significant increase with the onset of the Bronze Age (start of time period II) despite the fact that DNA content remained similar. Also, the power soil DNA kit has been successfully used in other sedimentary DNA studies spanning the late Holocene and in combination with downstream analyses like sequencing and qPCR (e.g. <https://doi.org/10.1128/AEM.02174-16> and <https://doi.org/10.1007/s10933-014-9807-1>). There is further support of our choice of extraction by the literature: In the recent review by Capo et al., (2021) (<https://www.mdpi.com/2571-550X/4/1/6/html>), it was proposed that the cyanobacteria 16S gene is currently the most-suitable marker in reconstructing cyanobacteria from ancient sediments. The same study also demonstrated that the sedaDNA of bacterial

groups were more efficiently detected and amplified by the same unmodified PowerSoil protocol used in the present study compared to intracellular DNA and extracellular DNA from a modified PowerSoil protocol. In addition, the unmodified PowerSoil kit for extracting total DNA showed a higher qPCR amplification success of DNA extracts obtained from bacteria groups (average quantification cycle (*C_q*) values were lower Capo et al., Fig. 5B, Case Study A4). In another previous study by Coolen et al., (2013)

<https://www.pnas.org/doi/full/10.1073/pnas.1219283110> up to ~560 bp in sediments spanning the last ca. 11,400 y. have been sequenced.

Nevertheless, we also acknowledge that there are limitations with kit-based extractions as you rightly mentioned which could partially bias our study, and we also think newly optimized hybridization capture baits technique like the PalaeoChip (Murchie et al., 2020) and (Armbrecht et al. 2021, <https://www.nature.com/articles/s41598-021-82578-6>) would result in even more efficient extraction of short DNA fragments and targeting of cyanobacteria sedaDNA. It needs to be considered though that these methods were optimized and tested against plant and animal DNA and marine eukaryote sedaDNA which contributes a much smaller proportion to total sedaDNA than prokaryotic sedaDNA. In the manuscript, we have addressed the possible limitations of our study and how aDNA fragmentation relates to our key result in the discussion section on lines 443-489. Also, we now replaced the word “robust” on the previous line 420 with “reliable”.

2) You didn't spend much time in the paper discussing why the cyanobacterial sedaDNA you're getting is actually ancient and not the result of some sort of contamination. Maybe I've missed it, but I didn't see a discussion of your blanks/negative controls. They don't seem to be included in any of your plots or tables, even though you say that you ran a blank for every 7 samples (L419-420). Judging by how many samples you have, you should have a lot of blanks by that ratio. Can you include an analysis of your negative control seq data to help demonstrate that these results aren't the result of contamination?

There were 11 extraction blanks in total, and DNA could not be quantified using Qubit from any of them. We PCR-amplified and sequenced two of the extraction blanks and two PCR negative controls (simply PCR blanks with only pure water). The DNA concentrations of the purified PCR products for both negative controls were low compared to those of the samples and positive control (see red columns and row in the attached NGS project table). On a gel (see attached gel electrophoresis image below), there was no band on the extraction blanks (pooled) and PCR blanks (pooled). The fact that DNA concentrations of the negative controls were low coupled with the absence of bands on gel electrophoresis, shows that external contamination of our samples can be neglected. After sequencing, the negative controls had few reads, which cannot be completely avoided in microbiome sequencing (see Hornung et al., 2019 <https://academic.oup.com/femsec/article/95/5/fiz045/5435435>) as these reads could either have resulted from the extraction kit 'Kitome' (see Salter et al., 2014 <https://bmcbiol.biomedcentral.com/articles/10.1186/s12915-014-0087-z>, de Goffau et al., 2018 <https://www.nature.com/articles/s41564-018-0202-y>), or index hopping during sequencing (e.g., MacConaill et al., 2018 <https://bmcbgenomics.biomedcentral.com/articles/10.1186/s12864-017-4428-5> and Sinha et al., 2017 <https://www.nature.com/articles/nbt.3981>). Considering that the samples were pooled into a library for sequencing, sequencing errors are expected to be the same across samples and should not bias our conclusions about the cyanobacteria population (Meyer et al 2012 <https://www.science.org/doi/10.1126/science.1224344>).

However, to remove any possible background noise from our samples, we subtracted the sum of the negative control sequencing reads from our samples before further downstream analysis. (see attached ASV table).

To make our negative control handling clear in the revised manuscript, we added the lines 655-658 "... and rare taxa were removed. The sum of the sequencing reads from the two extraction blanks and 2 PCR negative controls (simply PCR blanks with only pure water) were subtracted from the samples before further downstream analyses."

I'm surprised by how large of fragments you sequenced/targeted from the amplicon data, far exceeding what's typical of early-to-middle Holocene aDNA fragment length distributions.

We agree with the reviewer that this is a long fragment for Holocene and we are likely missing some target DNA by targeting a rather large amplicon fragment. However, as stated earlier, we have checked the length distribution of the extracted DNA via gel electrophoresis and, for older samples, even via tape station which revealed genomic DNA was sufficient for cyanobacteria-specific qPCR and amplicon analyses. As also already stated above, we targeted and sequenced ~359 bp cyanobacteria fragments based on recommendations from previous studies. For example, in Capo et al 2021 (<https://www.mdpi.com/2571-550X/4/1/6/html>) the cyanobacteria 16S gene (~400 bp) was proposed as appropriate in reconstructing cyanobacteria from ancient sediments. Also, Coolen et al., (2013) <https://www.pnas.org/doi/full/10.1073/pnas.1219283110> sequenced up to ~560 bp in sediments spanning the last ca. 11,400 y. Furthermore, the fragment size we targeted in this study (~359 bp) falls within the range (~100–500 bp) recommended by Boere et al., (2011) <https://www.sciencedirect.com/science/article/pii/S0146638011002269?via%3Dihub> as appropriate to be analyzed from Holocene sediments in order to minimize PCR bias caused by strand breaks.

Perhaps it's because they're from bacteria and they have sustained repair mechanisms that have stopped the fragments from getting too small. You sequenced quite deeply in the 14 shotgun samples. Deeply enough that I assume you could map your data to larger portions of cyanobacteria genomes than just 16S. Is it possible to do so in order to generate 1) mapDamage misincorporation plots to investigate deamination in your data (supporting that these sequences are ancient), and 2) fragment length distributions to look at rates of depurination?

Thank you very much for this constructive question. As suggested, we performed the ancient DNA damage analysis using two tools: mapDamage2.0 and pyDamage <https://peerj.com/articles/11845/>. To do this, we downloaded all available cyanobacteria genomes from NCBI and mapped all reads against it. As observed before, most reads mapped to *Cyanobium* species. However, we could not observe the typical “smile” pattern from the misincorporation plots. First, we thought this might be caused by the complexity of our samples with many very different species, and only a small fraction of reads mapping to cyanobacteria. After contacting the developers, we followed their suggestion and mapped against the most abundant single species only (*Cyanobium*), to avoid possible cross-mapping. Unfortunately, we still could not obtain any typical ancient DNA misincorporation patterns. This may have been caused by one or a combination of few reasons. Firstly, the commercially available library preparation kits with modified and improved proprietary protocols used by Eurofins (our commercial sequencing partner who cannot provide us with details due to property rights). We think the company uses Uracil Specific Excision Reagent (USER) or a UDG (Uracil-DNA Glycosylase) treatments during the library preparation. The USER and UDG treatments split the hairpin adapters, by generating a single nucleotide gap at the location of an Uracil. This removes the deamination signal and bias the library away from ancient molecules, thus no C-T or G-A misincorporations can be observed. Eurofins only says they use “TruSeq Adapters Sequences”. It is not clear whether they use Illumina’s TruSeq Adapters or the NEBNext adapters with TruSeq sequence. Another reason is the low coverage for cyanobacteria in the samples (between 10 reads in the deepest sediments and 13,000 reads in the topmost sediment). This coverage is too low for either all available cyanobacteria genomes from NCBI (1,119,256,536 reads) or the *Cyanobium* genome (3,342,364 reads). Lastly, the kit-based extraction may have also led to loss of shorter fragments that further make ancient DNA detection in downstream processes difficult. Nevertheless, we were able to obtain mean fragment length distribution from shotgun sequencing using mapDamage (shown below and the new Fig. 6 in the revised manuscript) for the samples which show cyanobacteria DNA fragmentation and reduction in the number of reads with increasing depth, and thus confirms that the cyanobacteria dynamics from the deep core were indeed ancient. In the revised manuscript, we have addressed all the above points on lines 443-472.

It would also be interesting (although not necessary if you think it's beyond the scope here) to use the approach from Kistler et al. 2017 (<https://doi.org/10.1093/nar/gkx361>) to look at how damage patterns from that shotgun data change through time relative to age/temperature. This data would help support that these cyanobacteria sequences are ancient, and ideally, that the amount of damage is roughly correlated with age, temperature, or other local factors influencing rates of aDNA damage. This would also help evaluate to what degree the shifting diversity you observe in cyanobacteria genera is more so related to changing intensities of human impacts or simply that the DNA isn't as degraded because it's younger (and hence you find more of it). These FLDs and/or depurination analyses will help with my previous question as to whether you may have lost more of your ancient DNA in the older samples, which could have artificially contributed towards the apparent increase in cyanobacteria DNA in the younger material.

We thank the reviewer for this suggestion, but we believe this is beyond the scope of the present study. The reason why we selectively sequenced some samples metagenomically was to confirm the dynamics we observed in the amplicon datasets, and not a systematic study based on metagenomic analysis alone. Moreover, as we showed in the results of the biomarker analysis (updated Fig. 2), the cyanobacteria lipid biomarker 7-methylheptadecane reconstruction also show similar variation as our cyanobacteria qPCR data, and *Aphanizomenon* ASV data from amplicon sequencing (updated Fig. 2). We have addressed the point of contamination above and are confident that it is not the source of cyanobacteria DNA in the sediment. We agree that DNA fragments must have shifted towards a higher proportion of shorter sizes deeper down in the core which potentially led to some failed detection through kit-based extraction and PCR approaches. Overall, based on the tape station analyses, the fragment lengths range in older samples was sufficient for cyanobacteria qPCR and amplicon analyses as mentioned earlier. In addition, if a systematic decay in DNA fragment size was the reason for the changes in qPCR data, we would have observed a gradual decrease in copy numbers. However, this is not the case, and we rather see a pronounced change from stable to steadily increasing numbers with the onset of the Bronze Age. This increase is clearly explained by a positive correlation to another paleo-proxy (pollen-based reconstructed vegetation openness) from the lake sediments and archaeological findings in and around the lake. Also, in the recent sedaDNA review paper by Capo et al 2021., the authenticity of sedaDNA older than 10,000 years and/or from poorer preservation settings can be demonstrated by identifying characteristic DNA damage patterns in metagenomic data. However, in our study, our key finding- sustained increase in cyanobacteria abundance at ca. 4,000 cal. a BP- is clearly supported by pollen-based reconstructed vegetation openness and archaeological findings in and around Lake Tiefer See. Based on these data we exclude contamination by modern DNA, now stated in the revised manuscript on lines 488-489.

It could also be interesting to run some phylogenetics with your mapped cyanobacteria shotgun data. This again might be beyond the scope here, but could be quite interesting with the data already available if you have the mapped coverages.

Thank you for the interesting suggestion. We agree that using the mapped shotgun data in phylogenetic analysis may reveal some exciting insights into DNA damage and/or evolution. This analysis is, however, beyond the scope of the present work and we, therefore, decided to refrain of it at this stage.

I realize this paper is focused on cyanobacteria, but I'm also curious what eukaryotic DNA was observed in the shotgun data? Can that be used at all to help support the claims that these shifts are driven by human activity (agricultural plants/animals showing up, human DNA, decrease in taxa susceptible to cyanobacterial toxins [like fish], etc.). This may be for another paper, but if not, it would strengthen your claims that these shifts are related to human activities and not climate. There are presumably many other organisms contributing to the changing ecology of this lake.

We fully agree that those data would provide even more support to our study. The assessment of eukaryotic composition has been addressed through additional shotgun sequencing and analyses conducted at the Alfred Wegener Institute in Potsdam. The data derived from these analyses are still under evaluation and final analyses need to be performed before data is ready for publication. For now, we can only confirm that the eukaryotic DNA from this shotgun data set seems promising in supporting our study but we cannot provide further information on it at this point which we hope you understand.

L147. The word concentration used twice, two words apart, consider changing to something like "The main finding is a distinctly higher sedaDNA concentration since..."

Thank you for picking up on this, the sentence was changed as suggested on line 157.

Fig 4. The caption doesn't make clear that this is from the shotgun data. It would also be nice if you included depths like in figure 3 so that the samples can be easily correlated with your supplementary tables.

Figure 4 is based on amplicon data, as written in the figure legend, and not shotgun data. We have now changed the caption to "Cyanobacteria Holocene beta-diversity analysis based on amplicon data" for clarity on line 214. The dots represent the sample depths analyzed. Including the depths of each sample would make the figure too crowded, hence the current color-coded cluster separation. In the figure legend we have now included "The dots represent the sample depths analyzed" on lines 1084-1085. We hope these changes make the figure clearer.

Sup Fig 1. It would be helpful to make the labels a bit more readable on this figure, similar to the main text map (which is beautiful by the way).

We have now replaced the figure with a new figure in which we made the labels, especially of the lakes and the place names, more legible, as well as made the symbols depicting the time periods larger. We hope these changes are satisfactory.

Great job! It was a pleasure to review this paper. Your manuscript has helped give me ideas on how to best analyze microbial sedaDNA data that I'm also working with.

-Tyler Murchie

Thank you once again for taking your time to review our manuscript, and providing such very helpful, insightful, and positively pleasant comments.

REVIEWERS' COMMENTS:

Reviewer #1 (Remarks to the Author):

The authors have addressed all my comments very well. Just be sure to specify consistently that the *nif* target was based on cyanobacterial specific *nif* gene (make clear in lines 674-681), since other microbial taxa can have the *nif* gene (expression being a very different question with respect to nitrogen fixation).

I have no additional comments. Well done!

Reviewer #2 (Remarks to the Author):

The revisions look good. My few remaining critiques are below. Line numbers refer to the newly revised document (the one without markup).

L420. Consider rephrasing slightly. As written now, it makes it seem like the USER treatment intentionally removes deamination and biases the library. Whereas because this kit is designed for modern DNA ultimately, the ancient DNA uracil problem is just due to this kit not being a good choice for multi-thousand-year-old DNA (it wasn't designed for this application). Also, UDG treatment isn't the same as USER, so line 419 isn't entirely accurate (UDG is for repairing miscoded uracils on the termini of insert DNA, whereas USER in this lib prep kit is designed for the hairpin adapters as you say).

I'd perhaps just say in line 420 something like:

"The USER treatment inadvertently removes deamination signals by excising uracil's expected on library hairpin adapters—uracil's which also happen to be present in ancient DNA. We would expect this to then bias a library away from ancient molecules..."

L428. The PalaeoChip baits are biotinylated RNA baits, it's just the (admittedly unnecessary) name that paper used to describe the faunal/floral arctic targets, so I'd rephrase here since this is like saying "...capture techniques like the PalaeoChip [apples] or biotinylated RNA baits [fruits]..."

L437. Why do you say this loss should be minimal? This TapeStation was done on extracts I assume? Did you use the D1000 or D5000 ScreenTape? If you used the D5000 (which is the more common one for genetics research), the typical resolution is 400-5000 bp, which means our group generally finds it's inaccurate for ancient DNA size ranges (we find it's typically infalted by ~100 bp compared to the D1000). Whereas, the D1000 has a typical resolution that can go down to 30-300 bp and is much more accurate with ancient DNA. Regardless, that tapestation is measuring total DNA, not cyanobacteria. And you said in the rebuttal that only a small portion of reads map to cyanobacteria. So that tapestation (if accurate) is really giving you the size range of a bunch of stuff that probably includes more modern contaminants as well (increasing the mean) that you didn't analyze. My critique here is that I'm not convinced you've demonstrated that the loss of small fragments is actually minimal, because you can't know what you've lost. This issue isn't unique here, but a bigger problem in the field (see Barta, Jodi Lynn, et al. "One of the key characteristics of ancient DNA, low copy number, may be a product of its extraction." *Journal of Archaeological Science* 46 (2014): 281-289).

I'd rephrase this and the next few sentences to be a bit more cautious. I also don't know what "(~100-500 bp) recommended for Holocene sediments" means by the way. What are the recommendations based? Most ancient DNA tends to fall within the 30-80 bp range, so 100-500 doesn't make sense to me. Anything over 100 is a small proportion of the distribution for a lot of ancient DNA research. See Figure 2 in "Marciniak, Stephanie, et al. "Ancient human genomics: the methodology behind reconstructing evolutionary pathways." *Journal of human evolution* 79 (2015):

21-34" for a fragment length distribution of what is pretty typical ancient DNA size ranges. I would think this sort of distribution applies here as well given that fragments are thousands of years old and reach into the early Holocene. My point is, that with >350 bp amplicons, non-ancient DNA optimized extraction/lib prep kits, and large-sized sequencing kits (2x150 bp instead of 2x100 or 2x50), which combined are all intended for more modern DNA (larger, undamaged stuff), there's the possibility of an undetected cyanobacteria sedaDNA signal in the early Holocene materials, but your methods didn't find it because it's too small given the known biases of the approaches you used. To be clear, I'm relatively convinced by the arguments you've made overall. I don't mean to be a stick in the mud. But this is a possibility that should be fully acknowledged in your limitations section.

Otherwise, though, I didn't spot anything else that I think needs changing beyond personal preferences that I won't trouble you with. You might consider removing the word 'reads' from each bar in Figure 6 and just putting that in the caption to clean it up a bit. But it doesn't really matter.

Nice work! Looking forward to seeing the published paper.

Reviewers comments

Reviewer #1 (Remarks to the Author):

The authors have addressed all my comments very well. Just be sure to specify consistently that the *nif* target was based on cyanobacterial specific *nif* gene (make clear in lines 674-681), since other microbial taxa can have the *nif* gene (expression being a very different question with respect to nitrogen fixation).

I have no additional comments. Well done!

Thank you for taking the time to read our revised manuscript once again. We have specified on the revised manuscript on lines 609 and 615 that the *nif* target was specific for cyanobacteria.

Reviewer #2 (Remarks to the Author):

The revisions look good. My few remaining critiques are below. Line numbers refer to the newly revised document (the one without markup).

L420. Consider rephrasing slightly. As written now, it makes it seem like the USER treatment intentionally removes deamination and biases the library. Whereas because this kit is designed for modern DNA ultimately, the ancient DNA uracil problem is just due to this kit not being a good choice for multi-thousand-year-old DNA (it wasn't designed for this application). Also, UDG treatment isn't the same as USER, so line 419 isn't entirely accurate (UDG is for repairing miscoded uracils on the termini of insert DNA, whereas USER in this lib prep kit is designed for the hairpin adapters as you say).

I'd perhaps just say in line 420 something like:

“The USER treatment inadvertently removes deamination signals by excising uracil's expected on library hairpin adapters—uracil's which also happen to be present in ancient DNA. We would expect this to then bias a library away from ancient molecules...”

Thank you, we have now rephrased the sentence as suggested on lines 408-410. We have also deleted UDG treatment from the revised manuscript.

L428. The PalaeoChip baits are biotinylated RNA baits, it's just the (admittedly unnecessary) name that paper used to describe the faunal/floral arctic targets, so I'd rephrase here since this is like saying “...capture techniques like the PalaeoChip [apples] or biotinylated RNA baits [fruits]....”

Thank you for your suggestion, we have now rephrased the sentence now on lines 418-421 to “... hybridization capture techniques like the biotinylated RNA baits (refs) to enrich for cyanobacteria aDNA. It needs to be considered though that the biotinylated RNA baits were optimized and tested against environmental and marine eukaryotic sedaDNA...”

L437. Why do you say this loss should be minimal? This TapeStation was done on extracts I assume? Did you use the D1000 or D5000 ScreenTape? If you used the D5000 (which is the

more common one for genetics research), the typical resolution is 400-5000 bp, which means our group generally finds it's inaccurate for ancient DNA size ranges (we find it's typically inflated by ~100 bp compared to the D1000). Whereas, the D1000 has a typical resolution that can go down to 30-300 bp and is much more accurate with ancient DNA. Regardless, that tapestation is measuring total DNA, not cyanobacteria. And you said in the rebuttal that only a small portion of reads map to cyanobacteria. So that tapestation (if accurate) is really giving you the size range of a bunch of stuff that probably includes more modern contaminants as well (increasing the mean) that you didn't analyze. My critique here is that I'm not convinced you've demonstrated that the loss of small fragments is actually minimal, because you can't know what you've lost. This issue isn't unique here, but a bigger problem in the field (see Barta, Jodi Lynn, et al. "One of the key characteristics of ancient DNA, low copy number, may be a product of its extraction." *Journal of Archaeological Science* 46 (2014): 281-289).

I'd rephrase this and the next few sentences to be a bit more cautious. I also don't know what "(~100–500 bp) recommended for Holocene sediments" means by the way. What are the recommendations based? Most ancient DNA tends to fall within the 30-80 bp range, so 100-500 doesn't make sense to me. Anything over 100 is a small proportion of the distribution for a lot of ancient DNA research. See Figure 2 in "Marciniak, Stephanie, et al. "Ancient human genomics: the methodology behind reconstructing evolutionary pathways." *Journal of human evolution* 79 (2015): 21-34" for a fragment length distribution of what is pretty typical ancient DNA size ranges. I would think this sort of distribution applies here as well given that fragments are thousands of years old and reach into the early Holocene. My point is, that with >350 bp amplicons, non-ancient DNA optimized extraction/lib prep kits, and large-sized sequencing kits (2x150 bp instead of 2x100 or 2x50), which combined are all intended for more modern DNA (larger, undamaged stuff), there's the possibility of an undetected cyanobacteria sedaDNA signal in the early Holocene materials, but your methods didn't find it because it's too small given the known biases of the approaches you used. To be clear, I'm relatively convinced by the arguments you've made overall. I don't mean to be a stick in the mud. But this is a possibility that should be fully acknowledged in your limitations section.

Thank you for bringing up this important concern which enables us to further acknowledge fully and explicitly the bias of our study on lines 392-397 "The DNA extraction and library preparation kits used were not optimized for ancient DNA, which might have led to the loss of some short and degraded DNA fragments, especially in the oldest sediment samples. That, and the use of relatively large targets (>350 bp) for amplicon and qPCR may have contributed to limiting the detection of the cyanobacterial signal in the early Holocene sediments, which is a potential bias of our study." We have decided to delete the lines on minimal fragment loss, because just like you said, we cannot be sure how many short DNA fragments were lost, either during extraction, library preparation, or sequencing.

Otherwise, though, I didn't spot anything else that I think needs changing beyond personal preferences that I won't trouble you with. You might consider removing the word 'reads' from each bar in Figure 6 and just putting that in the caption to clean it up a bit. But it doesn't really matter.

Thank you once again for taking your time to review our manuscript, and providing such very helpful, insightful, and positively pleasant comments. We have removed the word "reads" from each bar in Figure 6 and put it in the caption as suggested.

Nice work! Looking forward to seeing the published paper.